# A detailed kinetic model of Eastern equine encephalitis virus replication in a susceptible host cell

Caroline I. Larkin[1,2], Matthew D. Dunn[3], Jason E. Shoemaker[2,4], William B. Klimstra[3,5], James R. Faeder[2]*

**1** Joint Carnegie Mellon University - University of Pittsburgh PhD Program in Computational Biology, Pittsburgh, Pennsylvania, United States of America, **2** Department of Computational and Systems Biology, University of Pittsburgh, Pittsburgh, Pennsylvania, United States of America, **3** Center for Vaccine Research, University of Pittsburgh, Pittsburgh, Pennsylvania, United States of America, **4** Department of Chemical and Petroleum Engineering, University of Pittsburgh, Pittsburgh, Pennsylvania, United States of America, **5** Department of Immunology, University of Pittsburgh, Pittsburgh, Pennsylvania, United States of America

* faeder@pitt.edu

**Data availability statement:** Experimental data and code are available on the following GitHub

## Abstract

Eastern equine encephalitis virus (EEEV) is an arthropod-borne, positive-sense RNA alphavirus posing a substantial threat to public health. Unlike similar viruses such as SARS-CoV-2, EEEV replicates efficiently in neurons, producing progeny viral particles as soon as 3–4 hours post-infection. EEEV infection, which can cause severe encephalitis with a human mortality rate surpassing 30%, has no licensed, targeted therapies, leaving patients to rely on supportive care. Although the general characteristics of EEEV infection within the host cell are well-studied, it remains unclear how these interactions lead to rapid production of progeny viral particles, limiting development of antiviral therapies. Here, we present a novel rule-based model that describes attachment, entry, uncoating, replication, assembly, and export of both infectious virions and virus-like particles within mammalian cells. Additionally, it quantitatively characterizes host ribosome activity in EEEV replication via a model parameter defining ribosome density on viral RNA. To calibrate the model, we performed experiments to quantify viral RNA, protein, and infectious particle production during acute infection. We used Bayesian inference to calibrate the model, discovering in the process that an additional constraint was required to ensure consistency with previous experimental observations of a high ratio between the amounts of full-length positive-sense viral genome and negative-sense template strand. Overall, the model recapitulates the experimental data and predicts that EEEV rapidly concentrates host ribosomes densely on viral RNA. Dense packing of host ribosomes was determined to be critical to establishing the characteristic positive to negative RNA strand ratio because of its role in governing the kinetics of transcription. Sensitivity analysis identified viral transcription as the critical step for infectious particle production, making it a potential target for future therapeutic development.

repository: https://github.com/LarkinIt/EEEV-Replication-Model. Data generated from calibration and used for downstream analysis is available on Zenodo at http://doi.org/10.5281/zenodo.15080538.

**Funding:** C.I.L. was supported by NSF GRFP grant 1747452. J.E.S. acknowledges support from NIH NIAID grant R21AI153882, NSF CAREER #1943777, NIH NIAID 1R21AI174080, NIH NIAID R21AI151418, and NIH NIAID R01AI180382. W.B.K. acknowledges support from NIH NIAID grants R01AI160188 and R01AI153209. J.R.F. acknowledges support from NIH grants P41GM10371 and R01GM115805. Experimental studies were performed in the University of Pittsburgh Regional Biocontainment Laboratory, which is supported by NIH NIAID UC7AI180311. The funders had no role in study design, data collection and analysis, decision to publish, or preparation of the manuscript.

**Competing interests:** The authors of this manuscript have the following competing interests. W.B.K. is a co-founder of Advanced Virology. The Klimstra laboratory has received unrelated funding in support of sponsored research agreements from SAB Biotherapeutics. The remaining authors have declared no competing interest.

## Author summary

Eastern equine encephalitis virus (EEEV) is a positive-sense RNA virus transmitted via mosquitoes. It can cause lethal disease in humans with a high mortality rate, exceeding 30%. There are no licensed targeted treatments or vaccines currently available. We constructed a rule-based model that describes the mechanisms and the resulting dynamics of EEEV replication inside a mammalian cell. With a novel experimental dataset that measures the concentrations of EEEV RNA, proteins, and infectious viral particles over time in combination with a biological constraint based on known replication characteristics, we calibrated the model rate parameters with a Bayesian inference method that estimates parameter distributions and quantifies the confidence of model predictions. The resulting calibrated model captures key features of the experimental dataset. Model analyses identified a tight constraints in the RNA replication dynamics among the genome, the negative-sense template, and the subgenome, which is used for structural protein synthesis. The calibrated model demonstrates the potential for EEEV to rapidly recruit and densely pack host ribosomes on its viral RNA to accelerate replication. Sensitivity analysis found that parameters involving viral transcription, particularly of the genome and subgenome, are most critical for infectious viral particle production.

## Introduction

Eastern equine encephalitis virus (EEEV) is an arthropod-borne, neurovirulent virus endemic to the eastern seaboard of the United States. When neuroinvasive, it can cause severe disease characterized by meningitis and encephalitis with a human mortality rate between 30-70% [1]. Further, the overwhelming majority of survivors live with permanent neurological damage [2,3]. Currently, treatment is limited to supportive care as there are no preventative or targeted therapies. Additionally, EEEV can also be aerosolized, which poses a significant bioterror threat [4–8]. As of December 2024, an outbreak of Eastern equine encephalitis (EEE) in the eastern United States has generated 19 human cases across 8 states [9,10]. A previous outbreak in 2019 produced 34 symptomatic cases and 12 deaths [11–13].

EEEV is a single-stranded positive-sense RNA virus belonging to the Alphavirus genus in the *Togaviridae* family. Among alphaviruses, EEEV and the related Western equine encephalitis virus are unique in avoiding infection of lymphoid tissues due to restricted replication in hematopoietic cells [14–17]. EEEV replicates instead in osteoblasts, fibroblasts, and neurons by leveraging the efficient binding to the ubiquitously-expressed host proteoglycan, heparan sulfate, as well as several protein receptors [18–21]. The replication cycle is rapid, with viral particles being produced as early as 3-4 hours post-infection (hpi) [22] and cytopathic effects evident by 12 hpi [23]. The details of the EEEV replication cycle require further study because most previous studies of alphavirus replication were performed with laboratory strains of relatively avirulent arthritogenic alphaviruses, which likely differ from EEEV in key aspects. In this work, we develop a computational model of the dynamic replication strategy employed by EEEV in order to develop a better understanding of the mechanisms that contribute to the rapid timing and amplitude of particle production. This knowledge could help to guide the development of future therapeutic strategies.

Previous computational models have described intracellular replication in other positive-sense RNA viruses including Hepatitis C virus [24–30], SARS-CoV-2 [31–33], Dengue virus [30,34,35], and Coxsackieviruses [30,36]. However, despite its relevance as a human pathogen with limited treatment options, no computational model has been developed that

specifically addresses the replication cycle of EEEV or any other pathogenic alphavirus. Given experimental limitations, a computational model could be an invaluable tool to characterize the intracellular mechanistic underpinnings of EEEV replication.

Here, we present a detailed mechanistic model of the EEEV replication cycle that encodes alphavirus-specific mechanisms starting from particle attachment, culminating in the production and egress of both infectious and non-infectious progeny particles. This model contains detailed replication steps, including transcription, translation, and post-translational processing. Unlike previous models of viral replication in which ribosome density has been experimentally determined, we treat ribosome density as a variable parameter and use the length of RNA template strands to determine the polysome size (number of ribosomes in the translation complex) for translation. We estimate the value of this and other parameters by using Bayesian inference to calibrate the model based on experimental measurements of viral protein, RNA, and infectious particle dynamics during the early stages of infection. Using the calibrated model, we find that the recruitment and distribution of host ribosomes can play a critical role in the EEEV replication cycle onset and the resulting timing and magnitude of infectious particle production. Further analysis of the model suggests that the speed and regulation of RNA synthesis are the most critical determinants for the amount of infectious particles produced.

## Results

### A rule-based kinetic model of alphavirus replication

In order to obtain insights into alphavirus replication dynamics, we dissected the known replication steps into a series of discrete rate rules using the rule-based modeling language, BioNetGen [37] (Fig 1). Each biological process in the replication cycle is described by a rate rule based on the kinetic law of mass action. The remainder of this subsection presents an overview of the processes represented in the model. Additional details are provided in Materials and methods.

The first three model steps describe viral entry: attachment of an infectious viral particle to a host cell attachment factor in the cell membrane (Step 1), entry into the host cell (Step 2), and uncoating of the nucleocapsid to release the viral genome, which is then bound by host ribosomes (Step 3). After uncoating, it is assumed that the unbound host attachment factor is recycled to the cell surface (Step 4).

A unique aspect of positive-sense RNA viruses is the ability to form replication spherules (RS) using host-derived lipid membranes [38–41]. There are many potential functions of RS including concentration of the viral replication machinery, molecular trafficking of viral RNA and protein, and shielding from host immunosurveillance [42–45]. We incorporated RS into the model by including separate species for viral RNA located in the RS versus cytoplasm starting at Step 3.

With alphaviruses, the genome, denoted as posRNA in the model, is translated to make one of two possible nonstructural polyproteins (nsPs): P123 (Step 5) and P1234 (Step 6). These are differentiated by the presence or absence of nsP4, the alphaviral RNA-dependent RNA-polymerase. nsPs 1–3 serve numerous independent regulatory, replicative, and host-antagonizing functions but also synergize with nsP4 to create the replicase complex [46]. To catalyze RNA synthesis, nsP4 must be cleaved from the nascent polyprotein, P1234, but persists in a complex with the remaining polyprotein, denoted as P123-4 in Fig 1 (Step 7). This replicase complex subsequently binds to posRNA to form the replication complex RC:pos_neg (Step 8), from which negative template (negRNA) is transcribed (Step 9).

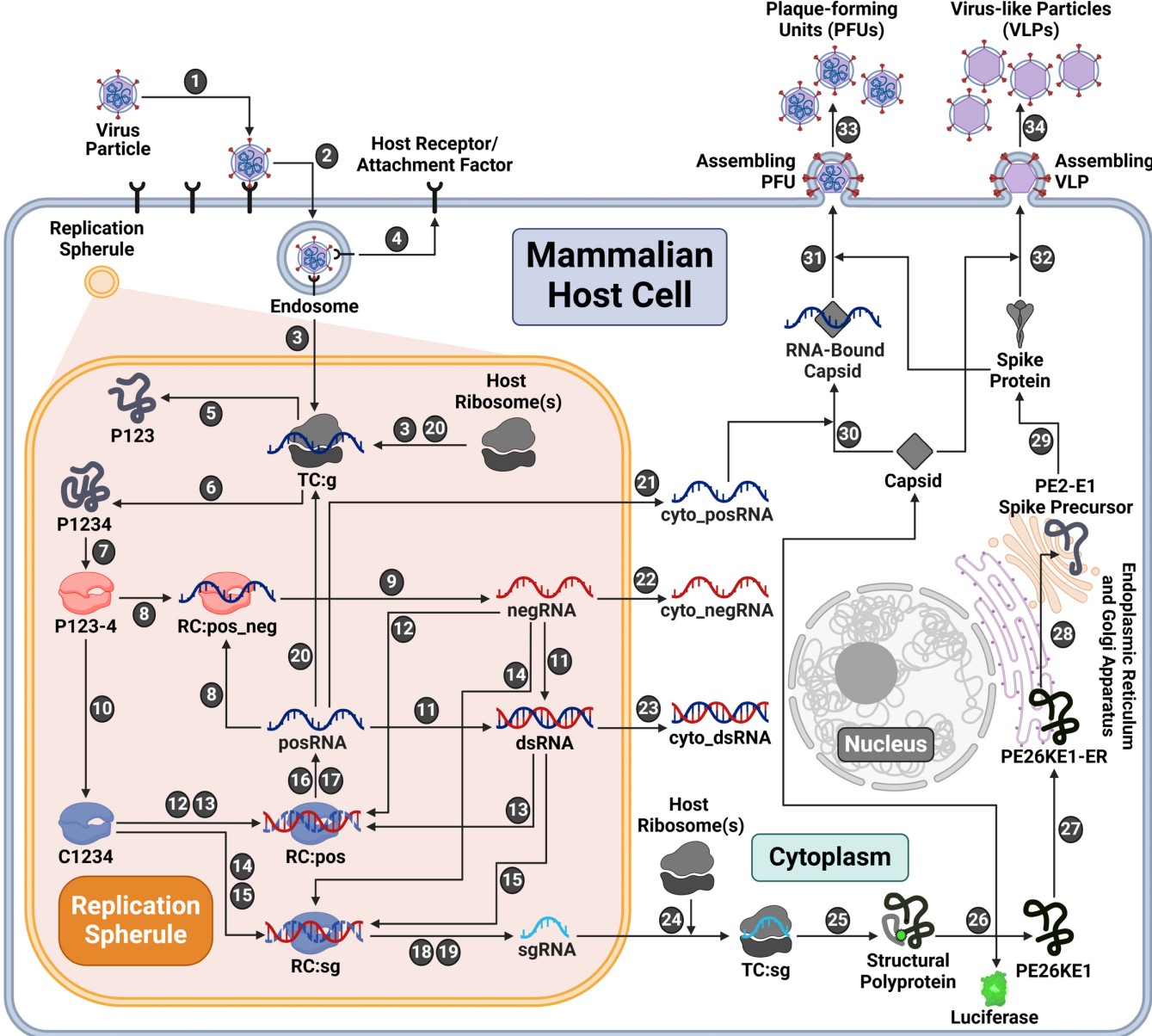

**Fig 1. Rule-based kinetic model of the EEEV replication cycle. Entry (Steps 1–4).** Infection begins with a viral particle binding to a host attachment factor to enter the cell. Subsequently, the viral envelope and capsid uncoat to release the viral genome (posRNA), which swiftly recruits host ribosomes to form the translation complex (TC:g). At the same time, the virus utilizes host lipid bilayers to form a replication spherule around replication machinery as shown with the orange-shaded region. **Replication (Steps 5–24).** Additional copies of viral RNA and proteins are synthesized. Host ribosomes synthesize the nonstructural polyproteins (P1234 and P123) from the genome and the structural polyprotein from the subgenome (sgRNA). To transcribe positive-sense RNA, a negative-sense template (negRNA) must first be synthesized by a competent replicase complex (P123-4). P123-4 is cleaved again into C1234, which catalyzes only positive-sense strand synthesis. Positive-sense RNA products are transcribed from either free or duplex forms of negative-sense strand template. There are two separate promoters on the negative-sense template strand that encode either posRNA or sgRNA, denoted as RC:pos or RC:sg respectively. sgRNA is bound by host ribosomes (TC:sg) to synthesize the structural polyprotein. **Structural polyprotein processing (Steps 25–29).** Once translated, the structural polyprotein undergoes a series of modifications through the secretory pathway yielding viral capsid and spike proteins. **Assembly and Export (Steps 30–34).** Before assembly, the capsid protein may bind to the genome. Both free and RNA-bound capsids assemble with spike proteins to form progeny particles. Plaque-forming units (PFUs) are infectious particles containing the infectious genome while virus-like particles (VLPs) are non-infectious due to the lack of a genome. Created with BioRender.com

A second proteolytic cleavage forms the late replicase complex, which is labeled C1234 (Step 10), which catalyzes only positive-sense strand synthesis. This irreversible step is critical for shifting transcription away from synthesis of the negative-sense RNA template and toward production of the positive-sense RNA products needed to produce progeny particles. Given their complementary nature, the posRNA and negRNA can pair to form double-stranded RNA (dsRNA) (Step 11). Both dsRNA and negRNA may be used as templates for positive-sense synthesis by the late replicase complex, C1234 (Steps 12–15). Depending on which promoter the replicase binds, these complexes may generate either progeny copies of the full-length genome (posRNA) (Steps 16-17) or the 26S subgenome (sgRNA) (Steps 18–19). The model assumes that viral RNA molecules synthesized in the RS translocate to the cytoplasm following first-order kinetics (Steps 21–23).

The requirement of host ribosomes to translate viral proteins has been established for a wide array of viruses [47–50]. Similar to eukaryotic translation, several copies of viral proteins may be translated simultaneously by multiple host ribosomes bound to a single strand of viral RNA, forming a complex called the polysome [48,50–53]. Alphaviruses such as Sindbis virus have been shown to employ polysomes for viral translation [54–56]. Other mechanistic models of positive-sense RNA virus replication have accounted for polysomes by fixing the composition based on virus-specific experimental observations [24,36]. Although polysome formation has not been studied with EEEV specifically, we assume that it does occur and incorporate polysome formation by the addition of a model parameter, `ribosome_distance`, that describes the average distance in nucleotides between ribosomes in a polysome on a single strand of RNA. The number of ribosomes in a polysome is then determined by the length of the RNA being translated divided by `ribosome_distance`. The polysome size thus differs by a factor of roughly three for translation of the genome and subgenome, which is approximately one third the length of the genome. We estimate `ribosome_distance` during model calibration (see next subsection) with bounds based on empirical measurements of eukaryotic translation [57]. The model assumes that the rate of translation scales linearly with the number of ribosomes in a polysome.

In the model, the fraction of host ribosomes available to EEEV is fixed at 20% (BNID: 102344, [58]), as assumed in existing models [36]. Host ribosomes can bind and form polysomes with free progeny copies of the viral genome (posRNA) to translate additional copies of the nsPs (Step 20) or move out of the RS (Step 21). Ribosomes also bind sgRNA to translate the structural polyprotein (Step 24–25). A nascent copy of the structural polyprotein undergoes several post-translational processing steps: auto-proteolytic cleavage of the capsid from the structural polyprotein (Step 26), translocation of the remaining polyprotein to the endoplasmic reticulum (Step 27), cleavage of the remaining polyprotein into individual proteins throughout the secretory pathway (Step 28), and cleavage, glycosylation, and palmitoylation of the spike precursors into mature spike proteins (Step 29) [59].

Functional viral particles, known as plaque-forming units (PFUs), form when capsid proteins bind first to cytoplasmic genome (Step 30), followed by assembly with the mature spike proteins into infectious particles (Step 31) that bud from the cell membrane (Step 33) (reviewed in [60,61]). The model assumption that a genome-bound capsid is required for PFU assembly is supported by studies with the Chikungunya alphavirus, which has shown that PFU production decreases when genome-binding sites on the capsid are mutated [62]. The model also allows capsid self-assembly to occur without prior binding of the genome (Step 32), resulting in formation of noninfectious virus-like particles (VLPs) that can then bud from the cell membrane (Step 34). Formation of VLPs via this mechanism is supported by experiments with Venezuelan equine encephalitis virus that have shown that noninfectious

VLPs are produced without the genome binding to the capsid, yielding genome-free particles [63]. Further, studies developing VLP-based vaccines for Eastern, Western, and Venezuelan equine encephalitis viruses have used vectors encoding the structural polyproteins for VLP production [64,65].

Overall, the model contains 36 species whose dynamics are defined with 60 rate rules. There are 36 model parameters, which include both initial conditions and rate constants for the individual replication steps (S1 Table). Of the 36 model parameters, 10 are fixed based on experimental observations (references in S1 Table). The remaining parameters are largely unknown, posing a significant obstacle to the biological relevance and predictive capability of the model. To obtain a better basis for estimating model parameters, we performed additional quantitative experiments to measure the concentrations of viral RNA, proteins, and infectious particle production in immune-incompetent fibroblast cells over the first 13 hours post-infection (hpi) with EEEV (see Materials and methods). We then used this data to carry out Bayesian inference of these unknown model parameters, as we now describe.

## Calibration of model parameters with Bayesian parameter estimation

Previous studies have shown that Bayesian methods can be effective in finding optimal model parameter sets that best fit experimental data from biological systems [66–68]. Unlike gradient-based optimization methods, which only return a single set of model parameters that fit the provided dataset, Bayesian methods return an ensemble of parameter set samples fitting the experimental data. Here, we used a recently-developed Bayesian method, preconditioned Monte Carlo (PMC), which combines Sequential Monte Carlo with Normalizing Flows (NFs) to improve the efficiency of moves in high-dimensional parameter spaces (see Materials and methods for details) [69]. We utilized PMC to estimate the 26 unknown model parameters of our EEEV replication model, as well as two experimental scaling factors, using the time course data of viral proteins, RNA, and progeny PFU production (Fig 2A).

When fit to the data without additional constraints, the model captures most features of the experimental dataset (Fig 2A, top row), including the timing and amplitude of the nonstructural (nsP) and structural (sP) polyprotein production (left), the synthesis of the genomic (posRNA) and subgenomic (sgRNA) RNA (middle), and PFU production (right). However, the predicted value of the RNA genome strand ratio (posRNA divided negRNA) is much less than one (Fig 2B, top row, middle panel), whereas some previous experiments with arthritogenic alphaviruses have found values in the range of 10–20 [70,71]. Using strand-specific RT-qPCR, we attempted to directly measure the kinetics of the negRNA, but this data was excluded due to a lack of sensitivity. We therefore sought to improve our model calibration strategy by incorporating an additional constraint to reflect a higher RNA genome strand ratio.

To constrain model behavior to be consistent with experimental observations, we implemented a penalty for parameter sets that produced a cumulative strand ratio below 20. Details of the implementation are provided in Materials and methods. Addition of the strand ratio constraint moderately increased the error in the fits (Fig 2B, left) while retaining most features of the experimentally-observed dynamics (Fig 2A and S2 Fig). The increased fit error arises largely from the model overshooting the observed posRNA (genome) concentration at 9 and 13 hpi (Fig 2A, middle). However, the constraint brings the predicted RNA genome strand ratio into the experimentally observed range as over 90% of the resulting ensemble has a cumulative ratio above 10 (Fig 2B). This shift in the strand ratio is achieved mostly through terminating negative-strand RNA synthesis by 5 hpi, when progeny particles are first produced, in conjunction with a modest increase in production of posRNA at the later times

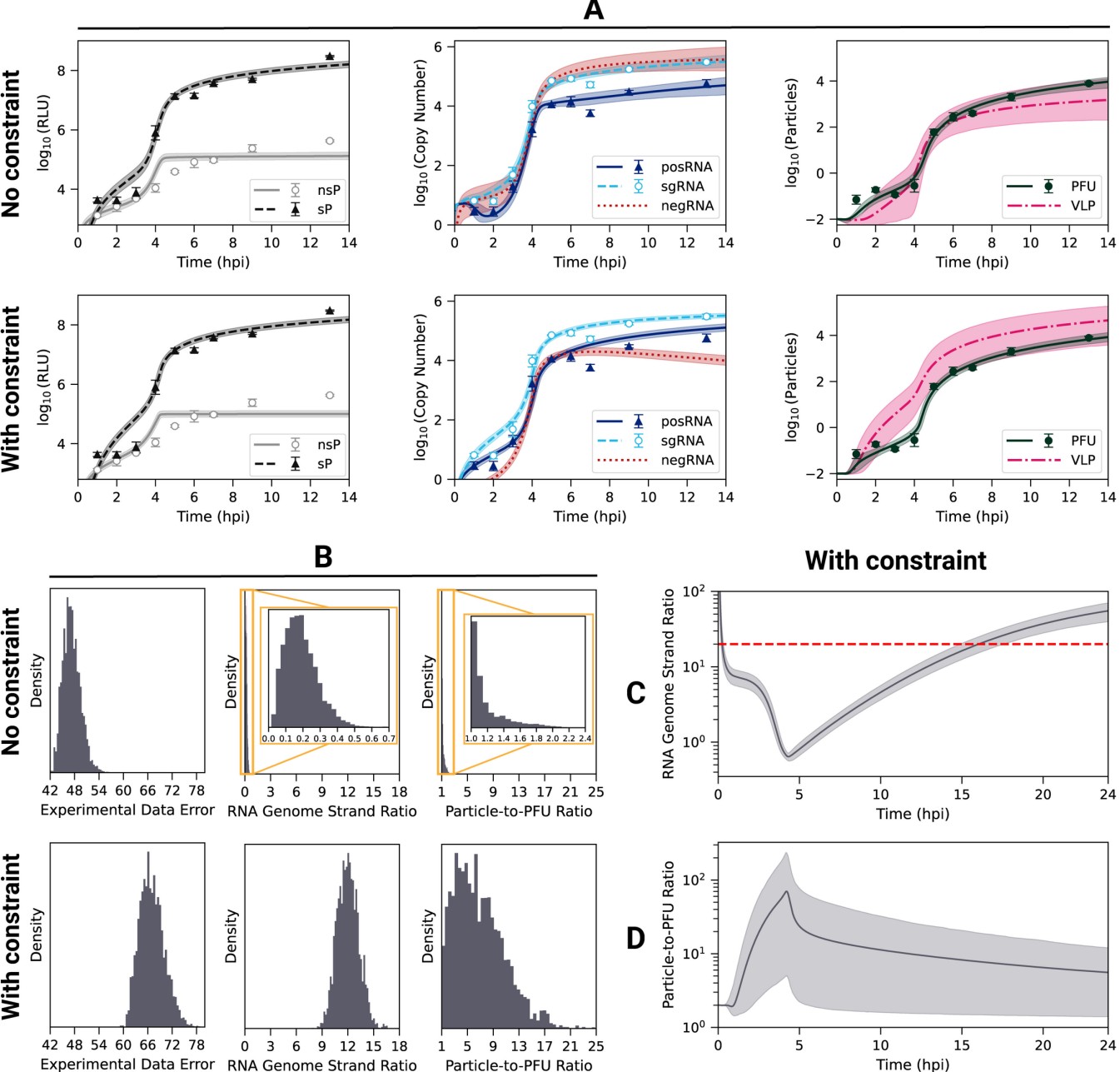

**Fig 2. Predicted and observed dynamics of EEEV replication following calibration with and without constrained RNA genome ratio.** (A) Time courses of (left) nonstructural (nsP) and structural (sP) polyproteins, (middle) RNA, and (right) infectious viral particles (PFU). The mean and 95% weighted credible interval of the model dynamics are plotted with the line and shaded regions, respectively. The mean and standard deviation of the experimental data are shown as symbols with error ($n = 3$ replicates). Experimental measurements of protein expression use polyprotein luminescence reported in relative light units (RLUs). (B) Histograms of the (left) experimental data error (a measure of the goodness-of-fit, defined in Materials and methods), (middle) cumulative RNA genome strand ratio (see text), and (right) cumulative particle-to-PFU ratio obtained (see text) from model calibration. The top row of panels A and B (labeled "No constraint") presents calibration results using only experimental data, whereas the bottom row (labeled "With constraint") shows results using both experimental data and the RNA genome strand ratio constraint. Model-predicted dynamics of the (C) RNA genome strand ratio and (D) particle-to-PFU ratio following calibration including the RNA genome strand ratio constraint. The red dashed line in C corresponds to the goal ratio used in the constraint. Histograms, means, and credible intervals are determined from a weighted ensemble of 2,500 parameter sets obtained from PMC.

(Fig 2A, middle). The predicted RNA genome strand ratio is highly dynamic throughout infection, exhibiting three phases (Fig 2C). The first stage occurs within the first hour as the infecting particles are binding, entering the cell, and uncoating to release posRNA, which results in a high strand ratio since there is no negRNA. From 1 to 4.5 hpi the ratio steeply declines as the viral replicase proteins are produced and begin negative-strand synthesis, causing the RNA genome strand ratio to fall briefly below one. negRNA dominance is short-lived, however, because its synthesis is shut off at about 4.5 hpi while the rate of posRNA synthesis increases, leading to the final phase of increasing strand ratio.

We next examined how the addition of the RNA genome strand ratio constraint affected the infectivity of progeny particles, which is quantified by the particle-to-PFU ratio: the total number of progeny particles divided by the number of PFU. When calibrating solely to the experimental data, PFU production generally outpaces VLP production (Fig 2A), resulting in a low average cumulative particle-to-PFU ratio of about 1.2 (Fig 2B, top row, right panel). The introduction of the RNA genome strand ratio constraint increases the average cumulative particle-to-PFU ratio to 6.9 (Fig 2B, bottom row, right panel), which is low compared to other positive-sense RNA viruses [72–75]. However, the low particle-to-PFU ratio is consistent with studies using alphaviruses that have been adapted to use heparan sulfate (HS) for entry [76–78], likely reflecting natural HS binding by EEEV [19]. Inclusion of the RNA strand ratio constraint increases the particle-to-PFU ratio by increasing the VLP assembly rate constant (Step 32, `r_vlp_assemble`) and decreasing the PFU assembly rate constant (Step 31, `r_assemble`) (S2 Fig). Together, these changes decrease the amount of posRNA packaged into PFUs, which increases levels of free posRNA in order to satisfy the RNA genome strand ratio constraint. Finally, the model predicts that the particle-to-PFU ratio peaks early during the course of infection and then decreases (Fig 2D), which is consistent with the experimental finding that infectivity increases over the course of infection with Sindbis alphavirus [76,79,80].

A notable benefit to Bayesian inference methods is the ability to quantify the uncertainty of model parameter values, often referred to as parameter identifiability, and model predictions using the posterior distributions of the estimated parameters, as we have already shown in Fig 2. The narrower the posterior distribution of a model parameter, the more identifiable it is. S1 Fig shows the individual (marginal) posterior distributions of the estimated parameters (26 model parameters and 2 scaling factors) across ten independent runs when calibrating with or without the RNA genome strand ratio constraint. In both cases the posterior distributions are well-converged across the 10 runs (S3 Fig and S4 Fig). The posterior distributions demonstrate that including RNA genome strand ratio constraint improves parameter identifiability, which is seen in the narrowing of the posterior distributions for a substantial number of the model parameters. The overall number of parameters varying over three orders of magnitude or more drops from 19 (68%) without the constraint to 14 (50%) with the constraint. Moreover, with the introduction of the RNA genome strand ratio constraint, several biologically important and potentially observable parameters including `r_endo`, `r_replicase_cleave`, and `ribosome_distance`, become well-constrained. Additionally, the vast majority of model parameters are not significantly correlated when using the RNA genome strand ratio constraint, as shown in S5 Fig. Taken together, the RNA genome strand ratio constraint effectively holds model behavior to be consistent with experimental observations while simultaneously improving parameter identifiability. From the resulting (posterior) ensemble of 2,500 parameter sets, we are able to identify and analyze robust behaviors of the model and characterize quantitatively key relationships between model parameters that arise from specific biological mechanisms.

## RNA strand synthesis is tightly regulated by the viral replicase

With a calibrated model that captures key characteristics of EEEV replication, we next focused on the kinetics of viral RNA synthesis, which is controlled by the viral replicases, P123-4 and C1234. Fig 3A shows that the calibrated model predicts replicase forms rise sharply between 3 and 4 hpi, but P123-4 (Fig 3A, red dashed line) declines steadily afterwards while C1234 reaches a high level and even continues to increase at a slower rate (Fig 3A, blue solid line). Except for a narrow window around 4–5 hpi, C1234 is by far the more prevalent replicase form, suggesting that the timing of negative strand synthesis is tightly regulated. The kinetics of the transcription fluxes (Fig 3B) reflect the dynamics of the replicase complexes. A surge in synthesis for all viral RNA occurs around 4 hpi, shortly after the jump in replicase concentrations. The production flux of negRNA briefly surpasses the posRNA production flux between 4–5 hpi but drops off rapidly afterwards. Moreover, within the same time window, the sgRNA becomes the dominant output of viral transcription (Fig 3B, light blue dashed line). Taken together, we see that negative-strand synthesis occurs within a narrow time window during which the intermediate P123-4 form of the replicase is prevalent. Afterwards, EEEV transcription switches to generation of positive-sense products in the form of posRNA, which can be directly packaged into viral particles, and sgRNA, which undergoes translation to produce structural proteins that make up viral particles.

We used the parameter set ensemble generated by PMC to determine whether the relative promoter strengths for production of genome (`ka_transcribe_4pos`) and subgenome (`ka_transcribe_4sg`) from negative-sense strand could be inferred from model calibration. The promoter affinities, which were independent parameters in the model, are not individually well constrained (S1 Fig), but their ratio is highly correlated (Fig 3C and S3 Fig, S5 Fig) with a mean value of about 5.1 $\pm$ 0.8 in favor of the subgenomic promoter (Fig 3D). This result suggests that the affinity of C1234 for the subgenomic promoter needs to be about five-fold higher than its affinity for the genomic promoter in order to generate the experimentally observed ratio of sgRNA to posRNA of about 2–10 (Fig 2A).

## EEEV allocates its genome for efficient assembly of infectious virions

A challenge specific to positive-sense RNA viruses is to maintain a distribution of posRNA to various replication processes in order to generate infectious particles efficiently, which are comprised of both transcriptional and translational products. posRNA is allocated over the following reaction fluxes: (1) *transcription* to produce nascent negRNA; (2) *translation* to produce the nsPs, P123 and P1234, which are required for viral transcription; and (3) *translocation* out of RS into cytoplasm for assembly into progeny PFUs.

To elucidate how model parameter estimates influence the distribution of the genome to various replication processes, we first examined the subcellular localization dynamics of the total genome in the cell (Fig 3E). In particular, genomes located in the RS correspond to *transcription* and *translation* while genomes in the cytoplasm indicate *translocation* for assembly. Following viral entry, the majority of the genome is located in the cytoplasm until 2.5 hpi. As the RS forms and the replication machinery is initially translated, genome allocation switches briefly to the RS as progeny copies of genomes begin to be synthesized around 4 hpi, but by 5 hpi the majority of genome copies has moved into the cytoplasm. By 10 hpi almost no genome copies remain in the RS.

Next, we investigated the specific fluxes of *transcription*, *translation*, and *translocation* as shown in Fig 3F. Prior to 4 hpi, the posRNA is primarily allocated to *translation* (Fig 3G, pink dashed curve). An abrupt increase in posRNA synthesis starting around 2 hpi (Fig 3B) drives increases in all three allocation fluxes. At about 4.5 hpi, *translation* drops abruptly due

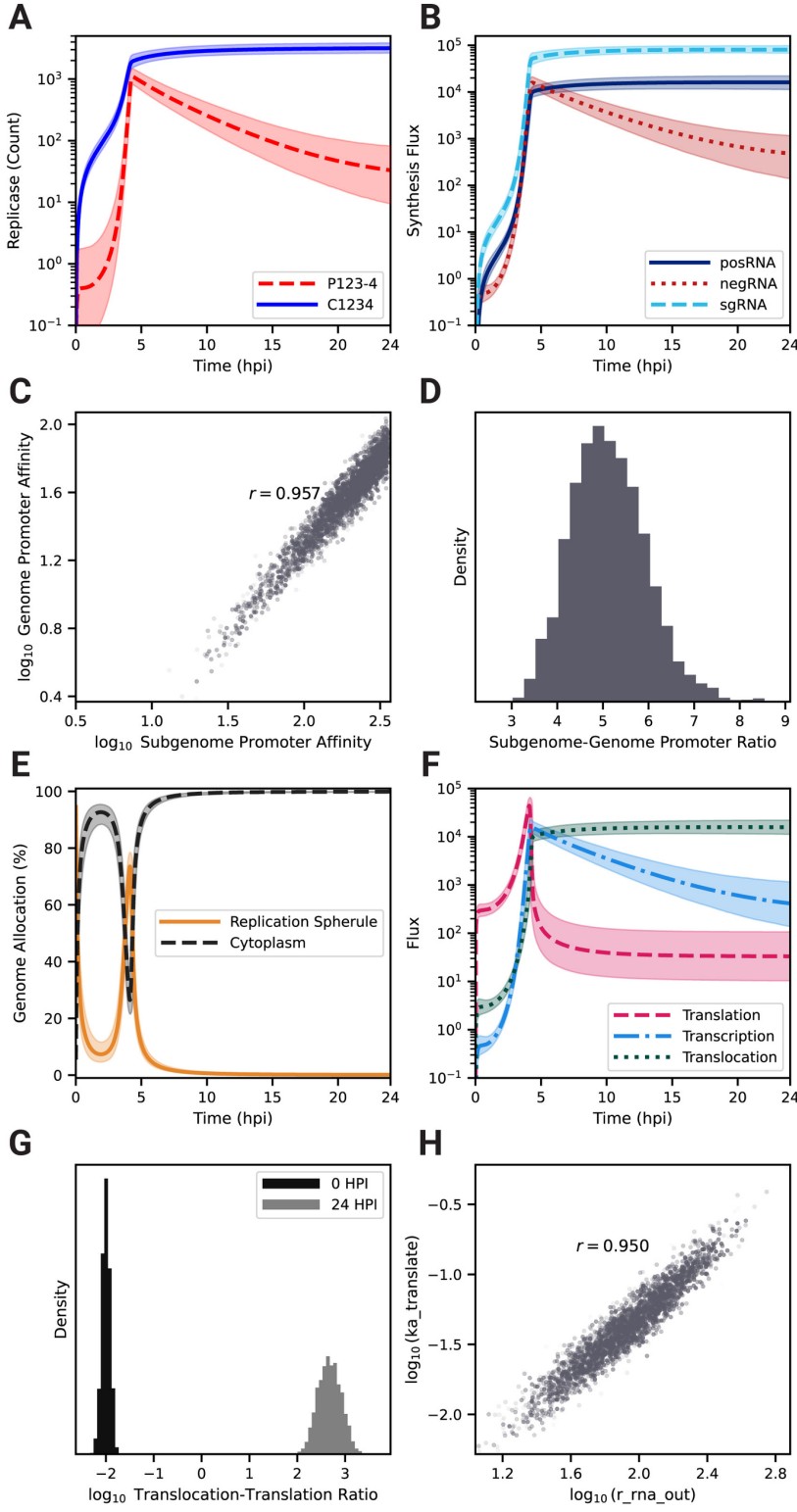

**Fig 3. EEEV employs a tightly-regulated genome replication and allocation strategy focused on diverting the genome for assembly into progeny viral particles.** (A) Model kinetics of the replicase complex at different functional stages: negative-sense competent (P123-4, dashed red) and positive-sense competent (C1234, solid blue). (B) RNA transcription fluxes for negRNA (red), posRNA (solid dark blue), and the sgRNA (dashed light blue). (C) Scatter plot of ensemble promoter affinities for the posRNA (`ka_transcribe_4pos`) and sgRNA

(`ka_transcribe_4sg`) sites on the negative strand and (D) histogram of the ratio (subgenome/genome) of these affinities. (E) Kinetics of fraction of genome located in replication spherules (solid orange) and cytoplasm (dashed black). (F) Kinetics of fluxes associated with viral genome for association with host ribosome (Step 20) (dashed pink), association with P123-4 (Step 8) (dash-dot light blue), and translocation out of the RS (Step 21) (dotted dark green). (G) Histograms of the ratio of translocation flux to ribosome association flux at 0 hpi (black) and 24 hpi (light gray). (H) Scatter plot of ensemble rate constants for genome translocation out of the replication spherule to the cytoplasm (Step 21, `r_rna_out`) versus the rate constant for translation initiation (Step 20, `ka_translate`), respectively. Histograms, means, and credible intervals are determined from a weighted ensemble of 2,500 parameter sets obtained from PMC. The transparency of individual samples in scatter plots is proportional to their corresponding importance weights. In Panels C and E the *r* value displayed is the Pearson correlation coefficient between the quantities shown on the *x* and *y* axes, respectively.

to decline of P123-4 (Fig 3B) but *translation* and *translocation* continue to rise. *Transcription* briefly dominates before dropping off, while *translocation* continues to increase and becomes dominant. The change of the *translocation* to *translation* ratio from very low early on to very high later during infection (Fig 3G) further demonstrates the dramatic nature of the shift that occurs. The strong correlation between the rate constants governing *translocation* and *translation* (Fig 3H and S3 Fig, S5 Fig) demonstrates that the balance between these processes must be tightly constrained in order to generate the observed viral replication dynamics.

To summarize, as replication is initially established, the genome is primarily used to synthesize the nonstructural polyproteins, but this allocation abruptly shifts around 4 hpi, when the genome is primarily translocated to the cytoplasm for assembly. Overall, there are three phases evident in Fig 3E–F describing the allocation of viral genome to replication processes: (1) translation-dominant; (2) transcription-dominant; (3) assembly-dominant. During phase (3) the sgRNA takes over the translation role for viral proteins, signaling a switch from nonstructural to structural protein production.

## High ribosome density on the viral genome is critical for rapid negative-sense template strand synthesis

Calibration of our model yields a posterior distribution of the distance between ribosomes on viral RNA (`ribosome_distance`) that is narrowly distributed about a mean value of 212 nucleotides (Fig 4C and S1 Fig), which is near the lower bound measured in eukaryotic cells [57]. This suggests that high ribosome density (i.e., low `ribosome_distance`) is necessary to achieve the experimentally observed replication kinetics (Fig 2). The average polysome compositions are calculated by dividing the respective RNA length by `ribosome_distance`. Thus, the model predicts that there are an average of 55 and 19 ribosomes bound to a strand of the posRNA and sgRNA, respectively. In the first 4 hpi, the available host ribosome pool does not exhibit substantial depletion, and posRNA and sgRNA do not compete for ribosomes. From 4 hpi onward, however, ribosomes undergo rapid depletion and bind predominantly to sgRNA (Fig 4A). Interestingly, translation of posRNA spikes sharply at 4 hours, falling off rapidly thereafter and mirroring its short-lived period of high concentration in the RS (Fig 3E). On the other hand, synthesis of the structural polyprotein using the sgRNA sharply increases until 4 hpi and is sustained for the duration of infection. Ultimately, it is rate limited by the total amount of host ribosomes available to the virus. Taken together, EEEV can rapidly utilize host ribosomes by densely packing ribosomes on the sgRNA to drive synthesis of the structural polyprotein and switch off the production of nsPs.

To further probe the relationship between host ribosomes and EEEV replication dynamics, we varied the parameters `Ribo_0` (Fig 4B), which is the fixed, initial pool of available ribosomes, and `ribosome_distance` (Fig 4C) to assess their influence on PFU production.

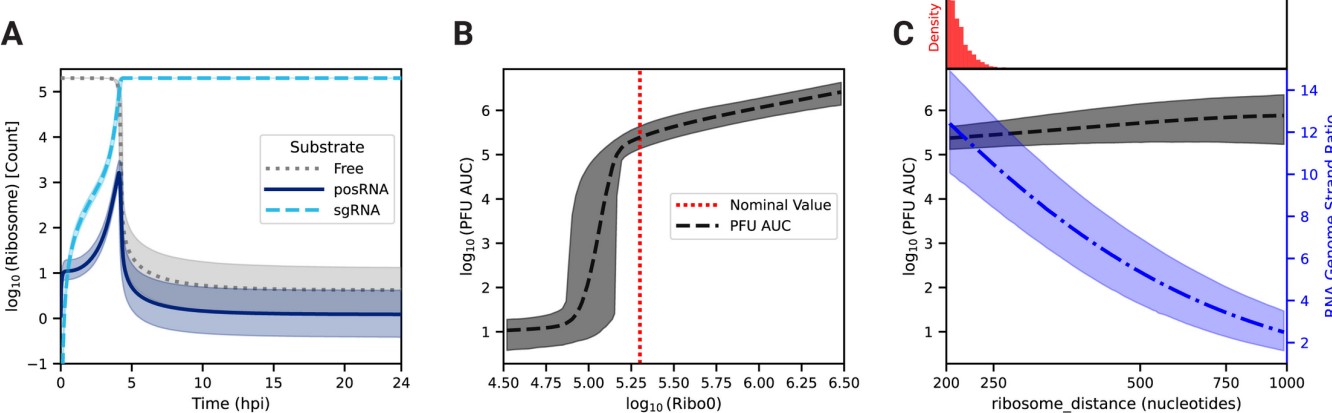

**Fig 4. Role of host ribosomes in viral replication kinetics and production.** (A) Free versus bound ribosome dynamics over 24 hpi. Ribosomes are in one of three possible states: free (dotted gray), genome-bound (posRNA, solid dark blue), or subgenome-bound (sgRNA, dashed light blue). Parameter scans of (B) the initial pool of ribosomes and (C) polysome composition as defined by `ribosome_distance` on infectious particle production (PFU, dashed black) and RNA genome strand ratio (dash-dot blue). PFU production is defined as the area under the respective species curve (AUC) after 24 hpi. For panel B, the vertical dotted red line represents the nominal value of the corresponding parameter (`Ribo_0 = 2 × 10^5`). For panel C, the estimated distribution of `ribosome_distance` after model calibration is shown in red. For all panels, the mean and 95% credible interval from the posterior samples are represented by the lines and shaded regions, respectively.

In order to quantify the uncertainty of our analysis, parameter scans of these two parameters were performed by varying the corresponding parameter over a range for each member of the parameter set ensemble. Fig 4B shows that the model predicts a sharp threshold of about $10^5$ available ribosomes is required for significant PFU production, which is just slightly lower than the fixed value used in the model calibration.

On the other hand, total PFU production is insensitive to polysome composition (Fig 4C, black dashed line), which is unexpected given how well-constrained the posterior distribution is for the corresponding parameter, `ribosome_distance`, after model calibration (Fig 4C, red bars). A further investigation revealed that `ribosome_distance` is primarily constrained by the RNA genome strand ratio, which remains within the experimental range only at the smallest values of `ribosome_distance` allowed (Fig 4C, blue dashed dot line).

To identify the origins of this behavior, we decreased the ribosome density by increasing `ribosome_distance` five fold over the parameter set ensemble. Decreasing the ribosome density in polysomes delays the time to host ribosome depletion by roughly 2-3 hours compared to the original model prediction, but, perhaps counterintuitively, lowers the final steady state number of free ribosomes (Fig 5A). posRNA production exhibits a similar delay in onset (Fig 5B), but becomes similar at later times. In contrast, decreased ribosome density generates increased steady state nsP levels (Fig 5C) by a factor of approximately 3. The increase of nsP production in turn allows for about an order of magnitude more negRNA to be generated (Fig 5D) compared to the original ribosome density, which explains the decrease in RNA genome strand ratio seen in Fig 4C. Fig 5E shows that sgRNA binding to ribosomes is significantly delayed, which expands the window of opportunity for the genome to be bound and translated by host ribosomes to produce nsPs and accounts for the increased production of nsP seen in Fig 5C. Fig 5F shows that PFU production is more modestly affected compared with the RNA genome strand ratio. In summary, decreased ribosome density in viral translation delays the kinetics of viral replication in the model through the interplay of transcriptional and translational mechanisms. Slower translation to produce replicase introduces a

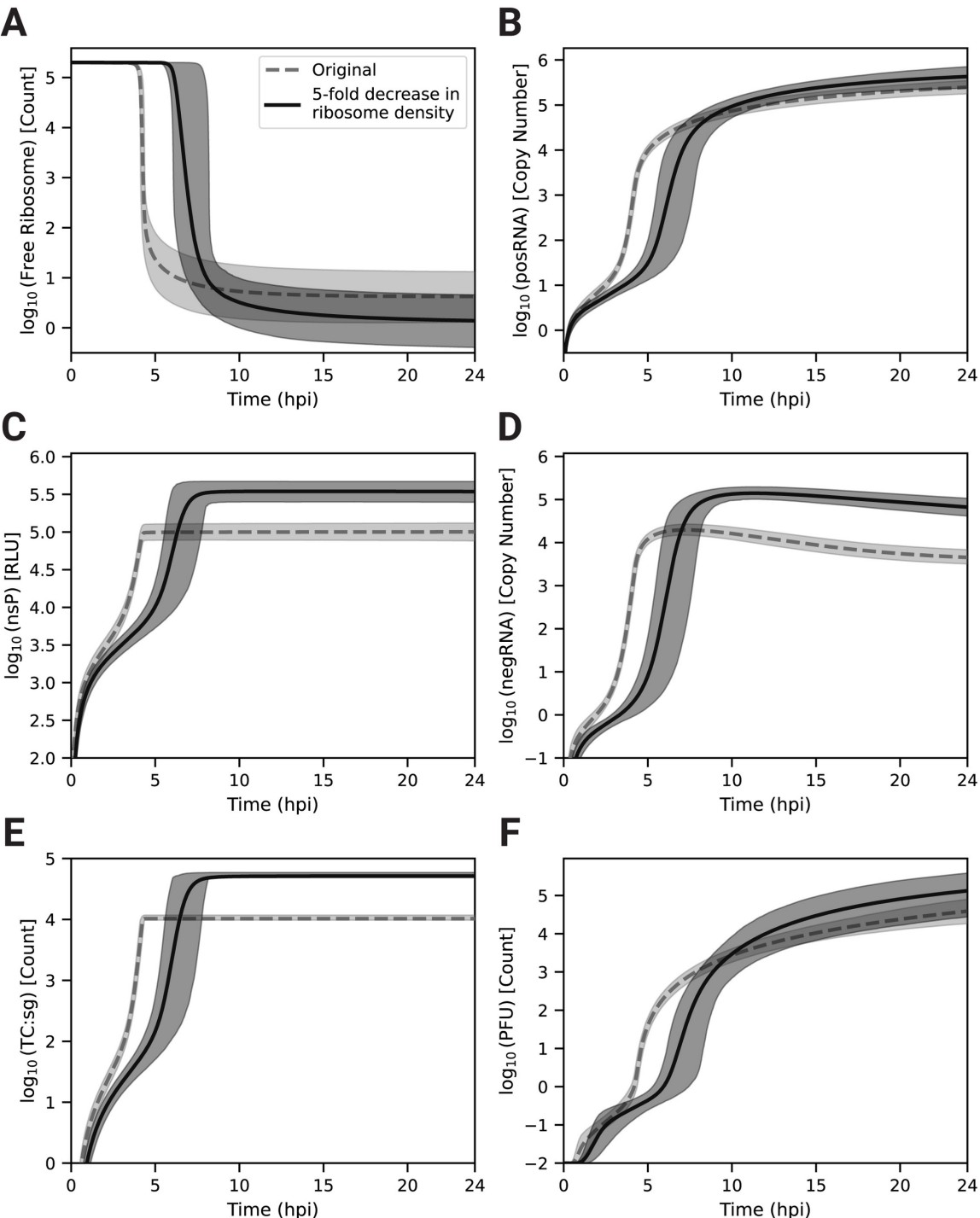

**Fig 5. The number of ribosomes bound to a single strand of viral RNA is a significant determinant of replication onset and magnitude.** The predicted dynamics of (A) free host ribosomes, (B) total genome (posRNA), (C) nonstructural polyprotein luminescence (nsP RLU), (D) negative-sense template (negRNA), (E) subgenome-bound polysome (TC:sg), and (F) infectious progeny particles (PFU) after increasing the average distance between ribosomes in a polysome (`ribosome_distance`) 5-fold over the parameter set ensemble. For all panels, the mean of the original trajectories and after a 5-fold increase in `ribosome_distance` are represented with solid and dashed lines, respectively. Shaded regions represent the 95% credible interval from the parameter set ensemble (*n* = 2,500).

delay in the kinetics of transcriptional products but also extends the window for translation of the genome to occur. This leads to generation of more nsPs and ultimately more negRNA, which accounts both for a moderate increase in the amount of PFU produced and a more dramatic decrease in the RNA genome strand ratio.

## One of three post-translational modifications to the structural polyprotein is rate-limiting

A potential benefit of Bayesian estimation methods is the ability to capture multiple regimes of parameter combinations that fit the experimental data to identify distinct possible mechanisms that can account for the observed dynamics. To search for such mechanisms, we performed hierarchical clustering on model species trajectories (see Materials and methods) and identified three distinct clusters of trajectories that do not differ significantly in their PFU production (Fig 6A and B) but differ in their distributions of the parameters describing the processing steps of the structural polyproteins through the secretory pathway, Steps 27-29 (Fig 6C–E). Each cluster is characterized by one of three rate parameters, `r_pro_translocate`, `r_pro_process`, and `r_pro_cleave`, being significantly lower in value and more tightly-constrained, while the remaining two parameters have higher values distributed over a broader range. The corresponding reactants can be distinguished by cluster (Fig 6F–H). Specifically, Clusters A (dark blue), B (orange), and C (light green) correspond to the build-up of either the structural polyprotein without capsid (PE26KE1), the translocated polyprotein in the endoplasmic reticulum (PE26KE1-ER), or the precursor spike proteins (PE2-E1), respectively. For each cluster, the concentrations of one of these model species reaches a steady-state level significantly and uniquely higher compared to the other clusters' respective species trajectories (Fig 6G–I). Taken together, this suggests that either structural polyprotein translocation to the endoplasmic reticulum, polyprotein post-translational modifications in the secretory pathway, or cleavage of the spike proteins is rate-limiting and critical for fitting the experimental data.

## Transcription speed and positive-sense promoter affinities are major factors driving infectious particle production

We next sought to identify critical model parameters that drive PFU production over the short infection period. For this, we employed a linear regression-based sensitivity analysis (LRSA) method (Fig 7A, see caption and Methods for additional details). Originally, we generated 100,000 parameter set samples using log-uniform Latin hypercube sampling of the model parameters from the model calibration bounds (see S1 Table). However, the vast majority of the sampled parameter sets (99.99%) resulted in PFU production levels below 100 particles, rendering this method impractical for further analysis (S6A Fig). As an alternative, we utilized the best-fitting parameter set from our parameter set ensemble as the reference point for sampling, varying each parameter by a factor of 10 in each direction. This approach results in the retention of 44% of the 10,000 generated samples after the filtering out samples that do not yield significant PFU production (Fig 7B and S6B Fig). We then performed linear regression to approximate the PFU production ($y$) as a linear function of the log parameter values ($\vec{x}$),

$$y(\vec{x}) = c_0 + \sum_{i=1}^{N} c_i x_i + \varepsilon, \tag{1}$$

where each coefficient $c_i$, estimated by linear regression, represents the linear sensitivity of the output $y$ with respect to the feature $x_i$, and $\varepsilon$ is the error.

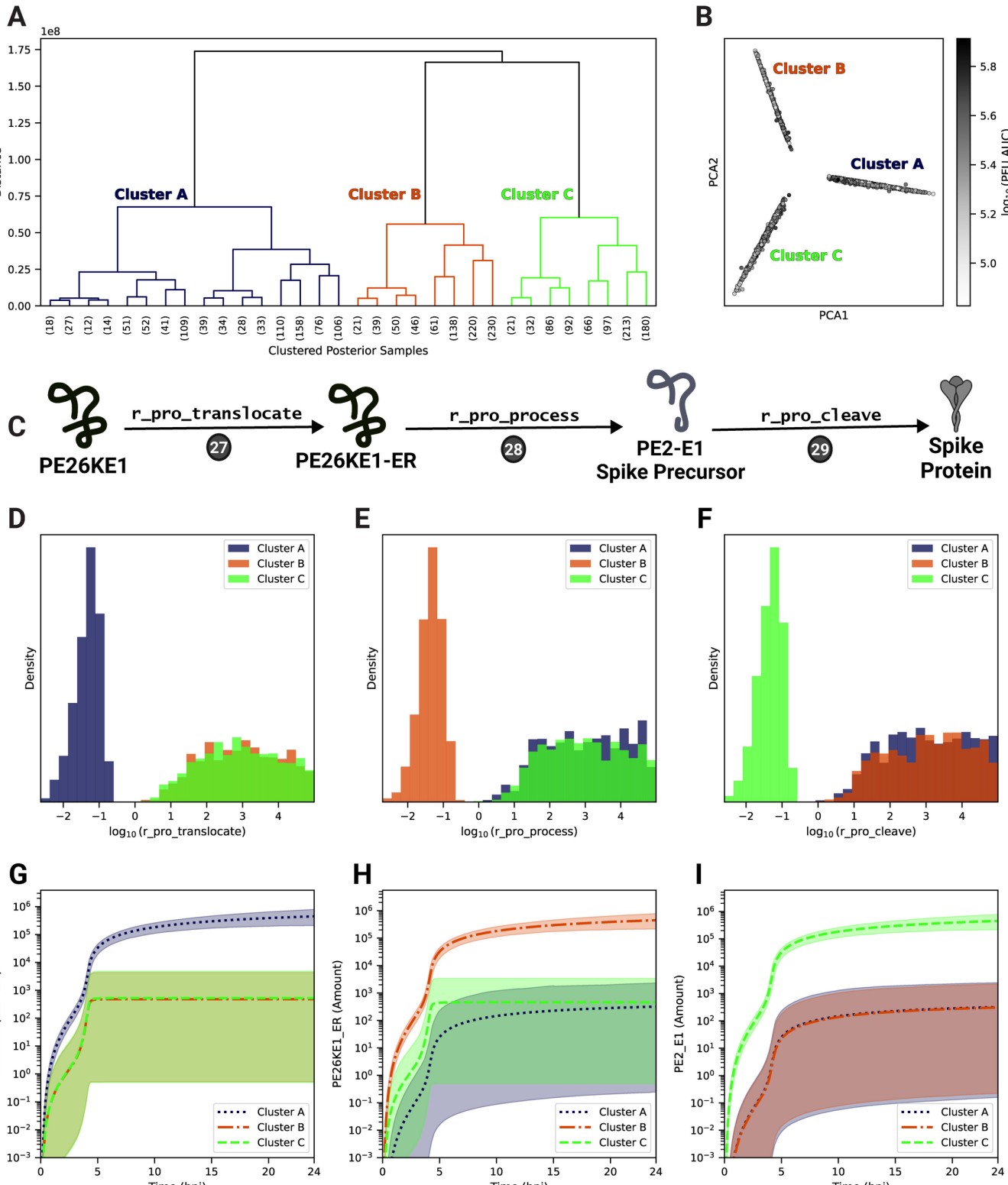

**Fig 6. Hierarchical clustering of trajectories from the parameter set ensemble.** (A) The estimated ensemble parameter sets (*n*=2,500) were hierarchically clustered using all model species trajectories over 24 hpi. (B) Principal components analysis of estimated parameter sets based on all model species trajectories. The shade of each sample corresponds to the PFU production (PFU AUC), which is defined as the area under the curve of the PFU trajectory.

(C) Schematic of the corresponding model rate rules (Steps 27-29) from Fig 1. (D)–(F) Estimated ensemble distributions, by cluster, of model parameters involved in post-translational protein processing: `r_pro_translocate` (Step 27), `r_pro_process` (Step 28), and `r_pro_cleave` (Step 29). (G)–(I) Cluster trajectories of the associated protein product of the corresponding rate rules with a rate shown in panels (D)–(F). Mean (line) and 95% credible intervals (shaded) for each cluster are plotted. Panel C was created using Biorender.com.

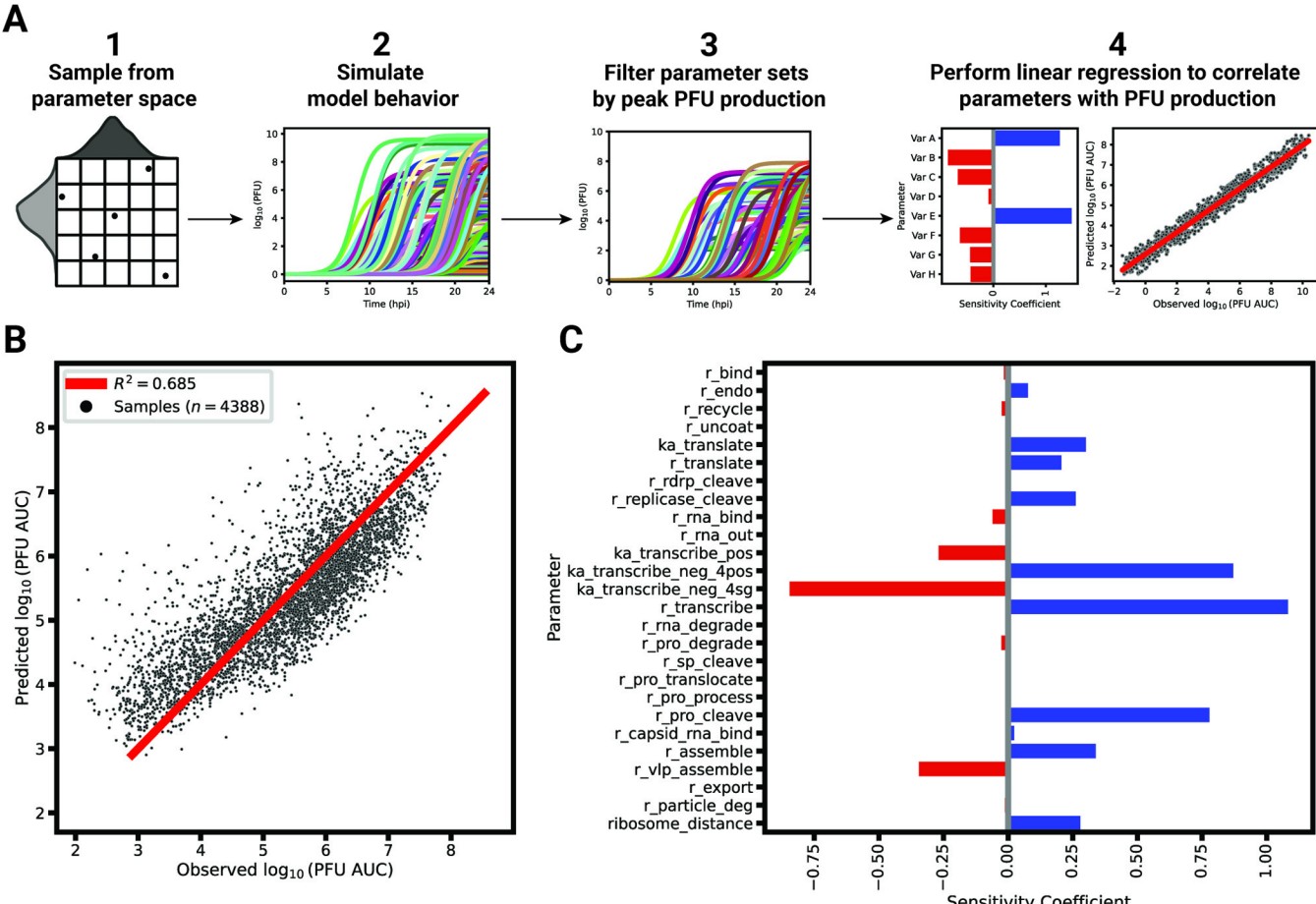

**Fig 7. Linear regression-based sensitivity analysis (LRSA) to identify model parameters most critical for PFU production.** (A) An overview of LRSA in the context of EEEV replication. To understand the relationship between the model parameters and viral production, (1) 10,000 parameter sets are generated and (2) subsequently simulated. The viral production for each sample is calculated as the area under the curve of the infectious particle trajectory. (3) Parameter sets yielding significant PFU production are selected. (4) The remaining parameter sets are used as features in a linear regression to predict their corresponding PFU production as defined by the PFU area under the curve (AUC) after 24 hpi. (B) Linear regression performance when fitting globally-sampled parameters sets to their respective PFU AUC. (C) The sensitivity coefficients of each model parameter ($n$ = 26). A positive (blue) sensitivity coefficient indicates a positive correlation with PFU production while a negative (red) coefficient indicates a negative relationship.

Overall, the linear regression model predicts PFU production with an $R^2$ = 0.685 (Fig 7B). Several parameters involved in transcription emerge as strongly affecting progeny PFU production as indicated by their sensitivity coefficients (Fig 7C). The parameter with the highest, positive sensitivity coefficient is the rate constant for transcription, `r_transcribe`. Other critical parameters include the relative affinities of the promoters on the negative-sense template for posRNA (`ka_transcribe_neg_4pos`) and sgRNA (`ka_transcribe_neg_4sg`), whose sensitivities have opposite signs. Since the subgenome promoter affinity has a negative coefficient while the genome promoter affinity

has a positive one, the sensitivity analysis suggests that a shift favoring posRNA synthesis relative to sgRNA will ultimately lead to increased PFU production. Based on the competition for ribosomes between the posRNA and sgRNA as shown in Figs 4 and 5, an increase in posRNA production relative to the sgRNA will increase nsP synthesis. By increasing the amount of nsPs produced, more negRNA is also synthesized before the replicases irreversibly shift to positive-sense RNA synthesis. Ultimately, this culminates in more posRNA synthesized from negRNA to be available to be packaged into progeny PFUs.

Another model parameter positively associated with PFU production is the processing rate of the structural polyprotein in the endoplasmic reticulum resulting in the precursor spike protein (`rna_pro_cleave`, Step 28). However, given the complex relationship with other structural polyprotein processing parameters, we sought to further probe the relationship between these correlated parameters with progeny PFU production. We performed LRSA for each of the three clusters identified in Fig 6 by globally sampling around the cluster's best posterior sample. The sensitivity coefficients for all model parameters other than the rate-limiting parameter remain similar among all clusters (S6C Fig). For each cluster, the rate-limiting parameter for that cluster (`r_pro_translocate`, `r_pro_cleave`, or `r_pro_process`) had the highest positive sensitivity coefficient, while the other non-rate limiting parameters had no significant relationship to PFU production. In summary, the LRSA results support the finding from Fig 6 that post-translational processing of the structural polyprotein contains a rate-limiting step critical for PFU production.

## Discussion

We constructed a mechanistic model of the Eastern equine encephalitis virus (EEEV) replication cycle within a mammalian host cell to better understand the replication dynamics resulting in the characteristic rapid production of progeny particles. We encoded the alphavirus-specific steps of intracellular replication into a series of interpretable rate rules starting with the attachment of a viral particle to a host attachment factor leading to the release of progeny particles differing in infectivity (Fig 1). Since the kinetic rate constants governing most of the processes considered in the model have not been directly measured, we utilized a Bayesian inference method to estimate them based on experimental measurements of EEEV infection kinetics during the first 13 hpi. To match experimental observations, we also incorporated a biological constraint into the model calibration process that favors synthesis of the positive-sense genome over the negative-sense template strand (i.e., a high RNA genome strand ratio). The resulting calibrated model captures key features of an experimental dataset generated from EEEV infection of mammalian fibroblasts. Further, the model suggests that the target RNA genome strand ratio of 20 is not reached until late in infection (Fig 2). In addition, the relative synthesis rates of the full-length genome, negative-sense template strand, and the subgenome are tightly-regulated by (1) a rapid transition of the replicase from its negative-sense to positive-sense transcribing forms, and (2) a constrained ratio between the genome and subgenome promoter affinities (Fig 3). Regression-based sensitivity analysis confirmed that the promoter affinities for the genome and subgenome strongly affect the amount of PFUs produced and also identified the importance of the overall transcription rate (Fig 7). Dynamics predicted by the model also reveal a constrained replication strategy that relies on allocating the majority of progeny copies of the genome to the cytoplasm for assembly. Finally, we found that the number of ribosomes in a single polysome is critical for establishing the RNA genome strand ratio, because a reduction of ribosome density leads to greater production of the negative-sense template strand and delayed production of infectious viral particles (Figs 4 and 5).

Initial calibration to the experimental dataset alone yielded an incorrect prediction, suggesting that the kinetics of the template strand followed those of subgenomic RNA, which reaches concentrations 2-fold or greater compared to the genome (Fig 2A). Constraining the model to match the experimentally-observed dominance of posRNA to negRNA only slightly decreased the quality of the fit to the experimental data (Fig 2A), while increasing the identifiability of a number of the model parameters, including `ribosome_distance`, which controls the number of ribosomes that simultaneously translate viral RNA. These results suggest the model is able to reproduce the observed kinetics of viral replication under the constraint of holding negative-strand RNA to low levels, which reflects empirical evidence and may be required for the virus to evade host immune responses [81]. The RNA genome strand ratio described by the model is highly dynamic as it sharply declines initially, briefly falling below 1 around 4 hours, and then rises rapidly thereafter (Fig 2C). Recent experiments measuring single-cell replication dynamics with the Sindbis alphavirus have similarly found that both positive and negative strands are synthesized at similar rates early in infection but shift towards positive-sense synthesis at later times [82].

Our model predicts that the termination of nonstructural polyprotein synthesis in combination with the rapid transition of the replicase form from negative-sense to positive-strand synthesis drives temporal variation of the RNA strand ratio dynamics. The model prediction suggesting that the replicase form responsible for negative-strand synthesis is short-lived is consistent with experimental observations in other alphaviruses [83]. Further, the model infers that negative-strand synthesis peaks at 4 hpi, which coincides with a rapid increase in the production of the positive-strand. The rapid increase in positive strand around 4 hpi has been recorded in the context of arthritogenic alphavirus infection [82–86]. Future experimental studies in combination with the incorporation of the innate immune response in the model, as described below, could help to further elucidate both the kinetics of and driving mechanisms for RNA strand asymmetry during alphavirus replication.

The model was constructed based on mechanisms applicable to any pathogenic alphavirus, and is thus easily adaptable through repeated model calibration. As a specific example, Semliki Forest Virus (SFV) lacks the opal termination codon between nsP3 and nsP4 [87]. As a result, SFV only produces the full-length nonstructural polyprotein (P1234), which can be modeled by setting a single model parameter (`f_nsp_scale` = 0) to zero. In a broader context, even though alphaviruses bind to many different attachment factors [88,89], the model structure would not need to change to accommodate these differences; only parametric changes including the initial amount of unbound receptors (`init_EU`) or the affinity of the virus particle to the respective host factor (`r_bind`) would be necessary. Thus, the model serves as a foundation to study the replication dynamics of any alphavirus that targets mammalian cells.

The model does, however, make a number of assumptions that might be productively addressed in future work. First, we describe transport of viral RNA from the replication spherule to the cytoplasm as a first-order kinetic process independent of any viral or host proteins. However, recent structural studies have shown that copies of nsP1 form a ring structure at the neck of replication spherules, suggesting a role for molecular trafficking of viral RNA [43,90,91].

Second, the model assumes that translation initiation is mechanistically and kinetically identical when using the genome or subgenome as the template. However, studies have shown that alphaviruses have developed mechanisms that accelerate translation initiation on the subgenome in order to circumvent host cell stress responses and viral shut-down of host translation (reviewed in [92]). Further, the intracellular location and timing of translation

may differ between the genome and subgenome, which may also affect ribosome recruitment [70,92].

Third, the competition for ribosomes between the host and virus is incompletely modeled. Generally, alphaviruses siphon off ribosomes from host translation using a two-pronged approach: (1) shut down of host transcription and translation and (2) recruitment of ribosomes to viral RNA [56,93–101]. Experiments investigating the subcellular location and composition of viral translation complexes would allow validation of model predictions such as high ribosome density on viral RNA and allocation of host ribosomes by the subgenome. Importantly, experiments could be used to test the feasibility of possible therapeutic strategies based on diminishing translation of the viral subgenome. In addition to host ribosomes, many other host factors play important roles in the propagation of alphavirus infection, including viral entry [88,102,103], RNA replication [104–108], and protein processing [109]. In the future, their inclusion in the model could help to elucidate the complex interplay between the host and virus that determine the outcome of infection.

In particular, the model currently simplifies structural polyprotein processing by consolidating the cleavage, folding, and trafficking of the structural proteins (except capsid) into a single step that does not include host proteins (Step 28). Additionally, the model does not explicitly include frameshifting of the 6K protein to produce the TF protein, which has been shown to play a role in alphavirus assembly [110–112]. Combined with experimental studies examining the dynamics of structural post-translational processing, further refinement of the model to include the details of these individual processes would provide the opportunity to determine specifically which post-translational mechanism is rate-limiting for infectious particle production.

Given the speed of EEEV replication, the adaptive immune response is often too slow to prevent infection from having catastrophic consequences for the host. Instead, the major determinant of acute EEEV infection outcome is the timing and amplitude of the innate immune response through the induction of Type I interferons (IFNs) [81,113–116]. EEEV and other alphaviruses have developed several mechanisms to antagonize the IFN response through the suppression of host translation and transcription, as well as other strategies [93, 113,117]. Although many of the mechanisms of interaction between the host and virus have been identified, the underlying dynamics and relationship to infection outcomes remain poorly understood, particularly in the context of EEEV infection. The model presented here serves as a starting point to probe host and viral antagonism through the future incorporation of the innate immune response.

## Materials and methods

### Mechanistic model of alphavirus replication

**Model construction.** The model is written in the BioNetGen Language (see S1 Table and S1 File) [37] and converted into the Systems Biology Markup Language Format (see S2 File) [118] for simulation using the RoadRunner Simulation Engine, which allows efficient compilation and integration of the ordinary differential equations generated by the model rules [119,120]. The model is composed of 60 mass-action rate rules and 36 individual species. For each reaction defined by a mass action rate rule, the rate of the reaction is given by the product of a rate constant parameter and the concentration of each reactant species. Overall, there are 36 model parameters: 10 are fixed based on published literature and 26 are estimated (S1 Table). Structurally, the model was designed to include replication mechanisms common in alphaviruses. However, given that the model is calibrated to EEEV data, we describe the model mechanisms and corresponding parameters below in the context of EEEV

infection. In the detailed description of the model processes below, the step numbers refer to those listed in the Rate Rules tab of S1 Table.

**Attachment, entry, and uncoating.**  Infection begins when a functional EEEV particle (`nV`) binds to an external unbound host attachment factor (`R_EU`) with rate constant `r_bind` (Step 1). We assume the initial number of viral particles is equal to the experimental multiplicity of infection (MOI), which is 5. EEEV particles predominantly bind to heparan sulfate, a ubiquitously-expressed proteoglycan [21], which we assume is in significant excess over viral particles by setting `R_EU` = $10^4$. The viral bound-host attachment factor (`R_EB`) enters the cell with a rate constant `r_endo` (Step 2) via a clathrin-mediated endocytic pathway, forming an intracellular endosome (`R_IB`) [89,121].

The next two rate rules encompasses multiple uncoating processes that serve to establish viral replication (Steps 3A and 3B). The four processes that occur during uncoating are: (A) endosomal compartment acidification, which leads to membrane fusion of the viral lipid envelope with the endosomal membrane and release of the nucleocapsid into the cytoplasm [122]; (B) nucleocapsid core disassembly to release the viral genome; (C) binding of viral genome by host ribosomes; and (D) formation of replication spherules (RS), which positive-sense RNA viruses employ to isolate and concentrate viral replication machinery through the rearrangement of host cell membranes [123]. Most alphaviruses including EEEV create these spherules using the lipid membrane from host endo-lysosomal membranes to form Type I-II cytopathic vacuoles [44,124,125]. The net reaction carried out by Steps 3A and 3B is the unbinding of the viral particle from host attachment factor to form unbound receptor (`R_IU`) and the genome translation complex (`translate_complex`, labeled TC:g in Fig 1). The process is split into two reactions in order to model the kinetics of polysome complexes that involve multiple ribosomes bound to a single viral genome, as described in more detail below. Although RS do not appear explicitly in the model species or rules, events described by rules 5–20 involve species that are confined to RS except for the host ribosomes and subgenomic RNA. In summary, processes A-D outlined above are combined into a single rate rule with a corresponding rate constant, `r_uncoat`. Lastly, unbound, intracellular heparan sulfate is recycled to the extracellular surface defined by the rate constant `r_recycle` (Step 4).

**Translation and polysomes.**  The EEEV genome (`posRNA`) and subgenome (`sgRNA`) are used as templates for translation of the nonstructural (`P123` and `P1234`) and structural (`s_poly`) polyproteins, respectively [54]. We assume that there is a fixed pool of free host ribosomes (`host_ribo`) available to the virus upon initial infection (BNID: 106861, [58]). An important component of translation regulation is polysome formation, in which multiple ribosomes bind to a single RNA strand to amplify protein synthesis. The number of ribosomes in a polysome has been shown to increase with the length of RNA being translated [57]. To capture this effect, we used a parameter, `ribosome_distance`, which represents the average distance in nucleotides between ribosomes on an RNA strand, to determine the average number of ribosomes in polysome complexes.

The model assumes that polysomes form in a single step, which is equivalent to assuming that all translation occurs from the steady state polysome complex as previously described in [24]. The net reaction to form a polysome is:

$$\text{RNA} + n\text{Ribo} \rightarrow \text{RNA}(\text{Ribo})_n \tag{2}$$

In BioNetGen the kinetics of polysome formation are split into two rules, which in generic form are:

```
A:  RNA + ribo -> TC + ribo  ka
```
$$\text{B:} \quad \text{RNA + ribo -> RNA} \quad \text{ka} * \frac{\text{length\_RNA}}{\text{ribosome\_distance}}$$

where `TC` represents the resulting translation complex, `ka` is the forward binding rate constant for ribosome–RNA binding, `length_RNA` is the RNA length in nucleotides, and `ribosome_distance` is the average distance in nucleotides between ribosomes bound to the viral RNA. Each rule generates differential equation terms affecting only species that undergo a net change in stoichiometry between reactants and products: `RNA` and `TC` in rule `A` and `ribo` in rule `B`. As a result, the differential equation terms arising from the these two rules are:

$$\frac{d[\text{TC}]}{dt} = -\frac{d[\text{RNA}]}{dt} = \text{ka} * [\text{RNA}] * [\text{ribo}]$$

$$\frac{d[\text{ribo}]}{dt} = \text{ka} * \frac{\text{length\_RNA}}{\text{ribosome\_distance}} * [\text{RNA}] * [\text{ribo}]$$

where square brackets indicate concentrations of the associated species. These rate rules and corresponding differential equations ensure that the translation complex species have the stoichiometries defined in Eq 2. Specific implementations of these generic polysome complex formation rules can be found in Steps 3, 20, and 24.

There are two nonstructural polyprotein products synthesized by host ribosomes using the genome: `P1234` and `P123`, differing in the synthesis of the RNA-dependent RNA-polymerase (nsP4). The resulting polyprotein product is determined by the readthrough of the opal codon between nsP3 and nsP4 [126]. For most alphaviruses, readthrough of this codon to produce `P1234` occurs 10-20% of the time [127]. This is reflected in the model using a fixed scaling factor, `f_nsp_scale`, which represents the frequency in which `P123` is synthesized. In summary, the rate of translation for `P123` (Step 5A) is defined as:

$$\text{translate\_P123} = \text{f\_nsp\_scale} * \frac{\text{length\_genome}}{\text{ribosome\_distance}} * \frac{\text{r\_translate}}{\text{length\_P123}} \quad (3)$$

and the rate of `P1234` synthesis (Step 6A) is:

$$\text{translate\_P1234} = (1 - \text{f\_nsp\_scale}) * \frac{\text{length\_genome}}{\text{ribosome\_distance}} * \frac{\text{r\_translate}}{\text{length\_P1234}}$$
$$(4)$$

In both functions, `r_translate` represents the individual ribosome translation rate (in amino acids $*$ h$^{-1}$), `length_genome` is the length of the genome in nucleotides, and `length_P123(4)` represents the length of the resulting protein product in amino acids.

Later in the replication cycle, the subgenomic RNA is used to synthesize a structural polyprotein (`s_poly`), which contains the individual proteins that comprise a progeny viral particle (Step 25). To maintain stoichiometric balance during particle assembly, the structural synthesis rate is scaled to reflect the number of structural proteins (`n_struct = 240`)

required to make a single viral particle [128,129]. Thus, translation of the structural polyprotein is defined by the following function:

$$\texttt{translate\_struct} = \frac{1}{\texttt{n\_struct}} * \frac{\texttt{length\_sg}}{\texttt{ribosome\_distance}} * \frac{\texttt{r\_translate}}{\texttt{length\_s\_poly}} \quad (5)$$

where `length_sg` is the length of the subgenome in nucleotides and `length_s_poly` is the length of the structural polyprotein in amino acids.

For all protein products, it is assumed that the rate of protein synthesis scales proportionally to the polysome composition and the polysome complex dissociates from the RNA template into individual ribosomes once synthesis is terminated. In BioNetGen, translation and turnover of polysomes is described by two rate rules, which in general form are:

$$\text{A:} \quad \texttt{TC -> RNA + protein} \quad \texttt{translate\_fun}$$

$$\text{B:} \quad \texttt{TC -> TC + ribo} \quad \texttt{translate\_fun} * \frac{\texttt{length\_RNA}}{\texttt{ribosome\_distance}}$$

where `translate_fun` is the translation rate function defined above (e.g., `translate_P123(4)` or `translate_struct`). Similar to translation initiation, these rate rules create corresponding stochiometrically balanced differential equations terms:

$$\frac{d\,[\texttt{protein}]}{dt} = \frac{d\,[\texttt{RNA}]}{dt} = -\frac{d\,[\texttt{TC}]}{dt} = \texttt{translate\_fun} * [\texttt{TC}]$$

$$\frac{d\,[\texttt{ribo}]}{dt} = \texttt{translate\_fun} * \frac{\texttt{length\_RNA}}{\texttt{ribosome\_distance}} * [\texttt{TC}]$$

The specific implementations of translation are defined in Steps 5, 6, and 25. Overall, the model assumes viral translation is localized to the replication spherule.

**Transcription.** In order for transcription to commence, the nsP4 must be cleaved off the polyprotein by the nsP2 with a rate constant, `r_rdrp_cleave` (Step 7) [130]. Although cleaved off, all 4 nsPs remain together comprising a replicase complex, as they all play regulatory roles in RNA synthesis [46,71,131–134]. The initial complex corresponding model species is denoted as `P123_4` (labeled P123-4 in Fig 1). This form of the replicase binds to the genome (`posRNA`) with a promoter affinity, `ka_transcribe_pos`, to form the transcription complex, `pos_replicase_4neg`, which is denoted as RC:pos_neg in Fig 1 (Step 8). This complex subsequently synthesizes the negative-sense RNA template, `negRNA` (Step 9).

A notable characteristic of alphaviruses is the further cleavage of the replicase complex which signals a shift from negative-sense synthesis to positive-sense synthesis [86,135–137]. Biologically, this shift occurs when first, the nsP1 is cleaved from nsPs2-3 and second, the junction between nsP2 and nsP3 is cleaved, leading to 4 individual proteins. The model combines these steps into a single rule (Step 10) because the intermediate form, where only nsP1 and nsP4 are cleaved off, is exceptionally short-lived [46]. The rate of this process is defined by the rate constant `r_replicase_cleave` and yields the replicase form responsible for positive-sense synthesis, `C1234`.

It is important to note that the model includes the binding of the free genome (`posRNA`) with the free, complementary template strand (`negRNA`) in the replication spherule, forming a double-stranded RNA intermediate, `dsRNA`, governed by the rate constant, `r_rna_bind`

(Step 11) [138]. Since the nsP2 in the replicase complex has helicase activity [139,140], we assume that the negative-strand in free or duplex forms can be used as a template to synthesize positive-sense RNA products. Specifically, `C1234` can initiate transcription through four mechanisms based on the form of the negative-sense template (free `negRNA` vs in-duplex `dsRNA`) and which of the two promoters it binds to. These promoters on the negative-strand correspond to the full-length genome (Steps 12, 13) and the subgenome, `sgRNA`, (Steps 14, 15). As a result there are four model species representing the possible transcription complexes: two for the genomic promoter, `neg_replicase_4pos` and `ds_replicase_4pos` (both denoted as RC:pos in Fig 1), and two for the subgenomic promoter, `neg_replicase_4sg`, and `ds_replicase_4sg` (both denoted as RC:sg in Fig 1). Previous studies, as well as our EEEV dataset, shown that the relative subgenome and genome production is asymmetric in favor of subgenome as the respective promoters are bound distinct sites on the RNA-dependent RNA polymerase [132,141]. We account for this in the model through the inclusion of two parameters, `ka_transcribe_neg_4pos` and `ka_transcribe_neg_4sg`, defining the genome and subgenome promoter affinities. We assume that these promoter affinities are not dependent on whether the negative-sense strand is free or in-duplex.

In general, we assume that only a single replicase is synthesizing a single strand of RNA at a time. Similar to translation, it is assumed that the transcription rate is dependent on the length of the viral RNA product. For full-length copies of the viral RNA (either `posRNA` or `negRNA`), the total rate is used for Steps 9, 16, 17 and is defined as:

$$\texttt{transcribe\_full} = \frac{\texttt{r\_transcribe}}{\texttt{length\_genome}} \tag{6}$$

and the subgenome transcription rate in Steps 18-19 is:

$$\texttt{transcribe\_sg} = \frac{\texttt{r\_transcribe}}{\texttt{length\_sg}} \tag{7}$$

where `r_transcribe` is the replicase transcription rate (in nucleotide $^{*}$ h$^{-1}$), and `length_genome` and `length_sg` are the lengths of the full-length RNA (`posRNA`,`negRNA`) and the subgenome in nucleotides. We assume that all transcription complexes (replicase + viral RNA) dissociate once the RNA product has been transcribed. Lastly, unbound forms of `posRNA`, `negRNA`, and `dsRNA` exit the replication spherule into the cytoplasm (labeled as `cyto_posRNA`, `cyto_negRNA`, and `cyto_dsRNA`) in a first-order rate rule governed by model parameter, `r_rna_out`, as shown in Steps 21-23.

**Structural polyprotein synthesis and processing.** Once synthesized, host ribosomes bind to the subgenome to form a translation complex (`sg_translate_complex`, labeled TC:g in Fig 1) in Step 24 and synthesize the structural polyprotein `s_poly` (Step 25) (see Translation section).

The following series of steps describes the processing of the structural polyprotein into functional individual proteins required for assembly into mature progeny particles [142]. First, the capsid protein auto-proteolytically cleaves itself from the structural protein (`s_poly`) yielding `capsid` protein, the remaining polyprotein, `PE26KE1`, and the nanoLuciferase reporter used to measure structural polyprotein production, `luciferase` (Step 26) [143,144]. The rate of Step 26 is defined with the rate constant, `r_sp_cleave`. The remaining polyprotein (`PE26KE1`) contains a signal sequence leading to its translocation to the endoplasmic reticulum, (`PE26KE1_ER`), determined by the

rate constant, `r_pro_translocate`, in Step 27 [145–147]. The polyprotein is transported through the secretory pathway towards the plasma membrane, undergoing a series of post-translational modifications which we model in a single rate rule governed by rate constant, `r_pro_process`, in Step 28 [70,122,147]. The resulting product is the spike precursor heterodimer, `PE2_E1`. pE2 on the heterodimer is subsequently cleaved by host protein furin to produce a fully mature spike protein [109,148]. We assume that the host protein furin is not rate-limiting, and thus assume this cleavage is a first-order reaction with a rate constant, `r_pro_cleave`, and yields the spike protein species, `E2_E1` (Step 29).

**Assembly and export of progeny viral particles.** The model proposes two mechanisms of assembly leading to two products: infectious, plaque-forming units (`PFU`) and non-infectious virus-like particles (`VLP`). In the model, the key difference between VLPs and PFUs is the presence of the viral genome. Mechanistically, the capsid protein preferentially binds to free genome in the cytoplasm (`posRNA_cyto`) yielding species, `rna_capsid` [62,149]. The corresponding rate is governed by the rate constant, `r_capsid_rna_bind`, as defined by Step 30. In the model, either free capsid or RNA-bound capsid will co-localize the spike proteins near the plasma membrane in preparation for the budding of progeny VLPs and PFUs, respectively. We assume this occurs at different rates for PFUs and VLPs as specified by `r_assemble` (Step 31) and `r_vlp_assemble` (Step 32), and ultimately yields the assembled model species `assembled` and `assembledVLP`. Lastly, the model assumes that both types of progeny particles bud out of the cell governed by the same rate constant `r_export` to produce progeny model species `PFU` and `VLP`, respectively (Steps 33, 34).

**RNA, protein and progeny particle degradation.** All free viral RNA, regardless of form (i.e., single-stranded vs duplex) or location (i.e., RS vs cytoplasm), undergo first-order degradation (Steps 35–41 in S1 Table) governed by model parameter `r_rna_degrade`. Similarly, all free viral proteins have first-order degradation (Steps 42–52 in S1 Table) defined by parameter `r_pro_degrade`. Extracellular PFUs and VLPs are assumed to degrade with an identical rate constant, `r_particle_deg` (Step 53–54 in S1 Table).

## Model calibration

**Processing experimental data.** In order to apply the experimental data, which is described below in the Experimental methods section, we converted the raw values of individual replicates ($n = 3$ per assay) into particles/units/copy number per cell from particles/units/number per mL using a cell density of $2.5 \times 10^5$ cells per mL. Additionally, we assume that since the MOI for all experimental data is 5, the number of virus particles available at $t = 0$ is 5. To account for the incubation period, we shift back all time points in the experimental dataset by one hour. Lastly, we exclude data collected after 12 hpi for model calibration. At 12 hpi, cytopathic events become evident [23], and since the model does not describe the effect of cell death on replication kinetics, the post-12 hpi data points are henceforth excluded. Additionally, we excluded the negative-sense EEEV RNA RT-qPCR data from model calibration due to a lack of sensitivity in the negative-sense RNA data from positive-sense RNA signal bleed over.

The model species, `PFU`, is directly fit to the plaque assay data. We assume that the genomic PCR dataset counts all forms of viral RNA with the genomic promoter. Thus we fit this dataset by summing the model species: `translate_complex`, `posRNA`, `pos_replicase_4neg`, `dsRNA`, `ds_replicase_4pos`, `ds_replicase_4sg`, `cyto_dsRNA`, `cyto_posRNA`, and `rna_capsid`. This is denoted as posRNA$_{ALL}$. To fit the

subgenomic PCR data, the following model species are summed with posRNA_ALL: `sg_RNA` and `sg_translate_complex`.

For the luciferase assays measuring the viral proteins, we calibrate an additional 2 parameters, `scale_nsp` and `scale_sp` to scale protein quantities per cell to relative light units (RLU). Specifically, to fit nsP3 luminescence data, all forms of the nonstructural polyprotein, free or bound, are assumed to be included in the assay. Moreover, the following model species are summed and subsequently multiplied by the scaling factor, `scale_nsp`: `P123`, `P1234`, `P123_4`, `C1234`, `pos_replicase_4neg`, `neg_replicase_4pos`, `neg_replicase_4sg`, `ds_replicase_4pos`, and `ds_replicase_4sg`. For the structural protein luminescence, we included a model species, `luciferase`, that represents structural polyprotein production as it is inserted and subsequently cleaved off as shown in Fig 1 Step 26.

To calibrate the model to the experimental data, we define the goodness of fit as follows:

$$C_{exp} = \sum_{i=1}^{n} \sum_{j=1}^{m} \frac{\left(\log_{10}(\bar{y_{ij}}) - \log_{10}(y_{ij})\right)^2}{\log_{10}\left(\sigma_{ij}\right)} \tag{8}$$

where $n = 9$ is the number of time points, $m \in \{\text{PFU}, \text{posRNA}_{\text{ALL}}, \text{sgRNA}_{\text{ALL}}, \text{nsP RLU}, \text{sP RLU}\}$. $\bar{y}$ is the mean experimental measurement and $\sigma$ is the standard deviation of species $j$ at time $i$ calculated from the triplicate data. $y$ represents the model output of species $j$ at time $i$.

**Integrating biological knowledge during model calibration with an additional constraint.** Given the complexity of the model calibration (26 model parameters and 2 scaling factors to be estimated), we implemented a constraint to be used with the experimental data set during model calibration. Previous studies of other alphaviruses have demonstrated that the synthesis of the negative-sense template maximally reaches 5% of its positive-sense counterpart [70,71]. Thus, we assume the fold-difference between the genome to template is 20-fold or greater. To integrate this during parameter estimation, we calculate the total genome and template produced over 24 hpi as follows:

$$Q_{RNA} = \frac{\int_0^{24} \text{posRNA}_{\text{ALL}} \cdot dt}{\int_0^{24} \text{negRNA}_{\text{ALL}} \cdot dt} \tag{9}$$

where posRNA_ALL and negRNA_ALL represent the total amount of genome and template strands, respectively. For the negative-strand, negRNA_ALL represents the sum of model species: `negRNA`, `cyto_negRNA`, `neg_replicase_4pos`, `neg_replicase_4sg`, `dsRNA`, `cyto_dsRNA`, `ds_replicase_4pos`, and `ds_replicase_4sg`.

Candidate parameter sets with a $Q_{RNA} < 20$ are assigned the following penalty:

$$H_{RNA} = \left(\log_{10}(Q_{RNA}) - \log_{10}(20)\right)^2 \tag{10}$$

The RNA genome strand ratio constraint is integrated with model fit to the experimental data to form a likelihood function as follows:

$$\mathcal{L} = -\left(C_{exp} + w_{RNA} \cdot H_{RNA}\right) \tag{11}$$

where $w_{RNA} = 10^2$ and represents the weight of the constraint error relative to the experimental data error. $w_{RNA}$ was manually-tuned from the following potential weights:

$w_{RNA} \in \{10^2, 10^3, 10^4, 10^5, 10^6, 10^7\}$. $w_{RNA} = 10^2$ resulting in the best balanced $Q_{RNA}$ and $C_{exp}$ values.

**Bounding parameters biologically.** All model parameters assume a log-uniform prior with bounds specified in S1 Table. To constrain parameters to stay biologically relevant during model calibration, we identified kinetic limits using a combination of experimental studies looking at binding kinetics as well as previously published mechanistic models of similar RNA viruses [36,57,150–155]. Numerous kinetic parameters, denoted as $X$, were provided in units, $M^{-1} \cdot s^{-1}$, and converted as follows:

$$X \, L \, mol^{-1} \, s^{-1} \cdot \frac{1 \, mol}{6.022 \times 10^{23} \, particles} \cdot \frac{3600 \, s}{1 \, h} \cdot \frac{0.001 \, m^3}{1 \, L} \cdot \frac{10^{18} \, \mu m^3}{1 \, m^3} \cdot \frac{1 \, cell}{1500 \, \mu m^3}$$

which yields a conversion factor = $3.985 \times 10^{-9}$. We assume the average cell volume is $1500 \, \mu m^3$ for BHK-21 cells [156,157]. The remaining unknown parameters are given wide bounds as specified in S1 Table. All calibrated parameters have log-uniform prior distributions based on the specified bounds.

**Parameter estimation using preconditioned Monte Carlo.** For all downstream model calibration and analysis, we used RoadRunner to load and simulate the model [119]. To calibrate the model, we utilized the Python implementation of preconditioned Monte Carlo (PMC), `pocoMC` [69]. PMC is an iterative algorithm that combines Sequential Monte Carlo and Normalizing Flows that repeatedly samples and evaluates candidate parameter sets from the initial prior distribution to the target, true distribution using importance sampling with Markov chain Monte Carlo. NFs are generative models that learn a bijective mapping of a multivariate distribution to a target distribution [158]. In particular, PMC is defined by the repetition of the following three steps:

1. (Re)weighing samples
2. Resampling
3. Mutation

PMC begins with a weighted ensemble of $N$ samples drawn from the prior distribution and then applies importance sampling to adjust the weights of each sample based on their likelihood. The importance weight for each sample is computed as the ratio of the target posterior distribution to the proposal distribution from which the samples were drawn. It is updated iteratively to better approximate the true posterior density. The updated importance weights, along with the sample weights from the previous iteration, are used to calculate new weights for each sample. Once the weights have been recalculated, PMC proceeds by resampling the ensemble based on these updated weights. Next, the mutation step evolves the ensemble of samples independently with MCMC-based methods, like Metropolis-Hastings, to allow for local exploration of the parameter space. To improve sampling efficiency, the PMC algorithm employs NFs to map the estimated parameter distributions from the ensemble to a multivariate Gaussian distribution. It is in this learned, Gaussian space in which the mutation step is performed.

Overall, this three-step sequence is repeated iteratively. However, unlike other MCMC methods, the number of iterations is not fixed. Instead, PMC uses a metric, denoted $\beta$, to assess convergence, and importantly, update the proposal distribution used for resampling. Specifically, the next $\beta_i$ is numerically estimated from the importance weights to maintain a user-specified fraction, $\alpha$, of independent samples from the ensemble (i.e., effective sample

size) (see Karamanis et al., 2022 [69] for details). PMC is considered to have "converged" once $\beta = 1$.

We use all default parameters when using `pocoMC` with the exception of the number of active particles and effective particles, which are both set to 2,500. The initial configurations of the particles is determined with Latin hypercube sampling of the model parameter bounds in log-uniform space. To assess convergence of PMC, the model was calibrated using 10 different initial configurations per calibration strategy (i.e., with or without the RNA genome strand ratio constraint).

## Model analysis

**Hierarchical clustering analysis.** We performed hierarchical clustering on posterior samples' trajectories for every model species over 24 hpi using the scikit-learn Python library [159]. These are then used to cluster posterior samples using the Ward linkage criterion. The number of clusters was manually assessed based on the dendrogram generated by hierarchical clustering. To examine how individual clusters contribute to the sensitivity analysis results, we additionally performed the regression-based sensitivity analysis on each individual cluster based on the best parameter set which is defined by the highest likelihood as specified in Eq 11.

**Linear regression-based sensitivity analysis.** To understand how the calibrated model parameters affect infectious particle (PFU) production, we employed global, linear regression sensitivity analysis. First, 10,000 parameter sets were sampled in log-uniform space using Latin hypercube sampling, yielding a $10,000 \times 26$ matrix. We do not include the experimental luminescence scaling factors in this analysis. To sample the parameter space effectively to yield any viral production, the best parameter set in the posterior samples was used to define sampling bounds. Moreover, these bounds are defined by multiplying the nominal parameter values from the best posterior sample by a factor of 10 in each direction. Model behavior is simulated to 24 hpi for each sample and the PFU production is calculated as follows:

$$\text{PFU}_{AUC} = \int_{t=0}^{24} \text{PFU} \cdot dt \tag{12}$$

Samples are subsequently filtered, keeping parameter sets leading to significant viral production, which is defined as a maximum PFU concentration between $10^2$ and $10^8$ within 24 hpi. Linear regression is performed on remaining samples using log-transformed parameter values as features and log-transformed $\text{PFU}_{AUC}$ as output. The corresponding weights from the linear regression are the sensitivity coefficients (see Eq 1).

## Experimental methods

**Materials availability.** cDNA clones are the property of Dr. William Klimstra and may be available upon request. Primer sequences and assay reagents are listed in S2 Table.

**Experimental model and reagents. Cell line**

**BHK-21 c**ells

The Baby hamster kidney (BHK) cell line is highly permissive due in part to defective interferon production. The BHK C-13 (ATCC CCL-10) cells were cultured in RPMI (Corning) 10% DBS (Gibco), 10% TBP (Moltox), 1mM L-Glutamine (Corning), 1x penicillin/streptomycin (Corning). Cells were infected in PBS (Corning), 1% DBS (Gibco) in a humidified incubator with 5% $CO_2$ at 37°C and overlaid with growth media + 0.05% immunodiffusion agarose (MP Bio).

**Virus stock**

**EEEV FL93-939 nLuc TaV**

The EEEV FL93-939, EEEV FL93-939 nLuc nsP3 fusion or EEEV FL93-939 nLuc TaV viruses were generated from a FL93-939 EEEV cDNA clone (provided by Scott Weaver, UTMB, Galveston) modified to express nanoLuciferase either as a non-cleaved fusion within the nonstructural protein 3 gene or as a cleavable fusion between the capsid and PE2 protein genes [160]. RNA genomes were synthesized in an *in vitro* transcription reaction with a T7 promoter RNA polymerase kit (Invitrogen) as previously described in [160]. Genomic RNA was electroporated into BHK-21 cells and viral supernatants were harvested and clarified 24 hours post electroporation followed by storage at -80°C. Viral titer was measured by BHK cell plaque assay (as in [160] and below).

**RT-qPCR standards.** Full length sense and negative-sense EEEV FL93-939 genomic RNA was generated from a dual promoter cDNA clone [7]. Transcription from the SP6 (Invitrogen) promoter yielded negative-sense full-length genome while transcription from the T7 promoter (Invitrogen) yielded positive sense full length genome. RNA genomes were treated with DNase (Invitrogen) and recovered with RNeasy Cleanup Kit (Qiagen) per manufacturer protocols. Concentrations were measured with a Nano-drop spectrophotometer and RNA was made into ten-fold dilution series to serve as RT-qPCR standards. Copy numbers for RNA genomes were calculated where $X$ = mass of amplicon and $N$ = length of amplicon using

$$\text{Copy Number} = \frac{X\,(\text{ng}) \cdot 6.0221 \times 10^{23}\,\frac{\text{molecules}}{\text{mol}}}{340 \times N\left(\frac{\text{g}}{\text{mol}}\right) \times 10^9\,\frac{\text{ng}}{\text{g}}}$$

**Virus growth curves.** BHK-21 cells were seeded into 24-well cell culture plates (Falcon) and allowed to grow overnight in standard growth media + 5% $CO_2$ at 37°C until they reached an average density of $2.4 \times 10^5$ cells per well. Cells were infected with EEEV viruses (0.1 mL infection medium) (MOI=5) for 1 hour at 37°C +5% $CO_2$ and brought up to 1 mL total volume in growth media after infection. Cells were washed 3x with cell growth medium prior to lysate harvesting. Viral lysate ($2.4 \times 10^5$ cells) and supernatant RNA samples (0.25 mL) were collected into Trizol reagent (0.75 mL) (Invitrogen). For plaque assay, supernatant fractions were collected (0.1 mL) in growth medium. For nLuc assay, cell lysates ($2.4 \times 10^5$ cells) were made in passive lysis buffer (0.1 mL) (Promega). Cells were reacted with virus in 0.1 mL infection medium for one hour, then washed 3x with growth medium. Samples were collected immediately after infection (experimental $t$=0) and again every hour for the first six hours, followed by collections at 8, 12, 16, 20, 24 and 36 hours post-infection. All samples were collected in triplicate from separate wells.

**Plaque assay.** BHK-21 cells were seeded into 6-well cell culture plates (Falcon) and allowed to grow overnight in standard growth media 37°C + 5% $CO_2$ until confluent. Cell monolayers were infected in duplicate with (0.2 mL) ten-fold dilutions of collected supernatant samples for one hour, 37°C + 5% $CO_2$. Cell monolayers were overlaid with 2 mL growth media + 0.05% agarose (MP Bio) and incubated 48 hours 37°C + 5% $CO_2$. Overlaid cells were stained with 2 mL neutral red solution (DPBS + 1% DBS, 1% Pen/Strep, 1x (Fisher) neutral red) overnight at 30°C + 5% $CO_2$. Plaques were visualized and counted over a standard light source.

**Protein nanoLuciferase assay.** NanoLuciferase protein expression was measured from triplicate collections of EEEV FL93 nLuc infected cell lysates by the manufacturer's protocol (Promega) as previously described in [160] on a Biotek Synergy LX plate reader.

**RT-qPCR assay.** Each 1 mL Trizol sample of infected cell supernatant or lysate was combined with 0.2 mL BCP (chloroform substitute) and 5 µL polyacryl carrier (MRC) and RNA extracted using a standard phenol/BCP (MRC) phase separation as previously described in [7] with an isopropanol precipitation and ethanol wash. cDNA synthesis was performed using the First-strand cDNA Synthesis M-MLV Reverse Transcriptase Kit (Invitrogen) protocol, including RNaseOUT (Invitrogen) [7].

To quantify the EEEV RNA genome, cDNA synthesis primers tagged with a T7 promoter (**bolded**) (IDT) were used to target the nsP2 coding region on both the positive and negative strands of the viral RNA.

Positive: 5'-**GCGTAATACGACTCACTATA**CACCGGCCAAAGTCTTCCATACTAT -3'.

Negative: 5'-**GCGTAATACGACTCACTATA**GCGCTACAAGGTCAATGAGAA-3'.

A similar strategy was used to quantify the EEEV RNA subgenome by targeting the capsid coding region on the positive strand with the following T7-tagged cDNA synthesis primer:

5'-**GCGTAATACGACTCACTATA**ACAGCTCCATGATGCCAGTTA-3'.

Thermocycling conditions were performed per manufacturer's suggested protocol (Invitrogen). cDNA was diluted 1:1 with nuclease free water (Hyclone) to 40 µL total volume and stored -80°C until further analysis. For quantification of genomic vRNA, qRT-PCR was performed using the 2x Fast TaqMan Universal PCR Master Mix, No AmpErase UNG (Applied Biosystems), following the manufacturer's instructions. Forward primer and probe targeting EEEV positive strand nsP2 include:

forward primer 5'-GCGCTACAAGGTCAATGAGA-3' and
probe 5'-ACGCACAGACATCTGAGCATGTGAA-3'.

Forward primer and probe targeting the negative strand include:

forward primer 5'-CACCGGCCAAAGTCTTCCATA and
probe 5'-TTCACATGCTCAGATGTCTGTGCGT-3'.

For quantification of the subgenomic vRNA, the following forward primer and probe were used to target the capsid gene:

forward 5'-GACCTTGAGTACGGTGATGTGC-3' and
probe 5'-GTACACCAGTGACAAGCCTCCTGGC-3'.

The probes were labeled at the 5' end with the reporter molecule 6-carboxyfluorescein (6-FAM) and quenched internally at a modified thymidine residue with Black Hole Quencher (BHQ1), with a modified 3' end to prevent probe extension by Taq polymerase. The T7 promoter tag introduced during cDNA synthesis was used as the reverse primer for each qPCR target: 5'-GCGTAATACGACTCACTATA-3'.

Thermocycler parameters consisted of initial denaturing: 95°C for 20 seconds and cycling PCR amplification (40 cycles): 95°C for 3 seconds and 60°C for 20 seconds. Quantification of viral genomes or subgenomes was determined by comparing the cycle threshold (CT) values

from unknown samples to CT values from the previously described 10-fold dilutions of *in vitro* transcribed sense and negative-sense EEEV RNA as described in [7].

## Supporting information

**S1 Table. Mechanistic model description including species, parameters, and rate rules.**
(XLSX)

**S2 Table. Key resources table.**
(XLSX)

**S1 Fig. Representative 1D marginal distributions for likelihood function ($\mathcal{L}$) and estimated parameters after calibrating the model with experimental data with and without the RNA genome strand ratio constraint.** Top left plot shows likelihood distribution as calculated by Eq 11. The remaining plots show 26 of the 36 model parameters in the BioNet-Gen model are calibrated as well as 2 scaling factors, `nsp_scale` and `sp_scale`, which are used to scale nonstructural and structural polyprotein concentrations to luminescence. These distributions are from the posterior ensemble of 2,500 parameter sets after calibrating the model given different set of initial particles. Specifically, the model was calibrated 10 times with the RNA genome strand constraint (blue) and 10 times without the constraint (orange). Vertical, dashed red lines indicate the parameter bounds provided during model calibration.
(TIF)

**S2 Fig. The model fits to the experimental data in linear space when performing parameter estimation with the RNA genome strand ratio constraint.** The posterior samples from model calibration ($n$ = 2,500) trajectory weighted means (line) and 95th percentiles (shaded) are shown in all panels. The mean and standard deviation of experimental data ($n$ = 3 replicates) are plotted from the protein luminescence assays, RT-qPCR, and plaque assay (see Materials and methods). Fits to protein luminescence data of nonstructural protein 3 (nsP) and structural protein (sP) in relative light units (RLUs) are shown in panels (A) and (B), respectively. (C) RNA dynamics of the positive-sense genome (posRNA, dark blue) and subgenomic RNA (sgRNA, light blue) are shown. (D) Model trajectories of viral particle production of plaque-forming, infectious particles (PFU).
(TIF)

**S3 Fig. The marginal posterior cumulative distributions of the likelihood and estimated parameters.** Model calibration to the RNA genome strand ratio constraint and experimental data was run 10 times given different initial configurations. The top left plot corresponds to the likelihood distributions (see Eq 11). The remaining 28 panels correspond to the 26 estimated model parameters and 2 scaling factors.
(TIF)

**S4 Fig. Pairwise Komolgorov-Smirnov statistic distributions for all model parameters across 10 independent runs of model calibration.** Using the resulting marginal cumulative distributions generated by each run of model calibration, as shown in S3 Fig, the two-way weighted Komolgorov-Smirnov statistic between each combination of runs (10 runs, 45 pairs for comparison) for the likelihood (labeled Total Error and defined in Eq 11), 26 model parameters, and 2 scaling factors. The black lines in each violin plot represent individual KS statistic for a pair of runs.
(TIF)

**S5 Fig. Pairwise calibrated parameter correlations.** The Pearson correlation coefficients between calibrated parameters ($n = 28$, 26 model parameters and 2 scaling factors) are calculated from the parameter set ensemble ($n = 2{,}500$).
(TIF)

**S6 Fig. Linear regression sensitivity analysis sampling approaches and results based on the clusters identified in Fig 6.** (A) Sorted distribution of maximal PFU yield from parameter set samples generated from global sampling with calibration bounds specified in S1 Table. (B) Sorted distribution of maximal PFU yields of parameter set samples generated using the best posterior set sample as described in Materials and methods. The number of parameter set samples generated are $10^5$ and $10^4$ for panels A and B, respectively. Green shaded regions in panels A and B correspond to the filtered PFU count range for samples to be used for the linear regression. (C) Cluster-based linear regression sensitivity analysis reveals three modes of post-translational processing dynamics. The sensitivity coefficients when sampling around the best posterior parameter set from each individual cluster ($n = 3$) using the same method described in Fig 7A. As shown in Fig 6, the left plot corresponds to the cluster in which `r_pro_translocate` is the rate-limiting parameter. The center shows the resulting sensitivity coefficients when sampling parameters around the best sample cluster in which `r_pro_process` is rate-limiting. The right plot corresponding to the sensitivity coefficients for model parameters using the best parameter set from the cluster in which `r_pro_cleave` is rate-limiting.
(TIF)

**S1 File. The mechanistic model in the BioNetGen Language format.** Remove the `.txt` extension before running with BioNetGen.
(TXT)

**S2 File. The mechanistic model in the Systems Biology Markup Language format.**
(XML)

## Acknowledgments

We thank the members of the Faeder research group as well as Rachel Gottschalk for useful feedback and discussions. Additionally, the first author would like to thank her canine companion, Clover, for her assistance throughout the course of this study.

## Author contributions

**Conceptualization:** Caroline I. Larkin, Jason E. Shoemaker, William B. Klimstra, James R. Faeder.

**Formal analysis:** Caroline I. Larkin, James R. Faeder.

**Funding acquisition:** Caroline I. Larkin, Jason E. Shoemaker, William B. Klimstra, James R. Faeder.

**Investigation:** Matthew D. Dunn, William B. Klimstra.

**Methodology:** Caroline I. Larkin, William B. Klimstra, James R. Faeder.

**Resources:** William B. Klimstra, James R. Faeder.

**Software:** Caroline I. Larkin, James R. Faeder.

**Supervision:** Caroline I. Larkin, William B. Klimstra, James R. Faeder.

**Validation:** Caroline I. Larkin, James R. Faeder.

**Visualization:** Caroline I. Larkin, James R. Faeder.

**Writing – original draft:** Caroline I. Larkin, Matthew D. Dunn, James R. Faeder.

**Writing – review & editing:** Caroline I. Larkin, Jason E. Shoemaker, William B. Klimstra, James R. Faeder.

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
