## [Decision Letter · Decision Letter 0]

17 Feb 2025

PCOMPBIOL-D-24-02237

A detailed kinetic model of Eastern equine encephalitis virus replication in a susceptible host cell

PLOS Computational Biology

Dear Dr. Faeder,

Thank you for submitting your manuscript to PLOS Computational Biology. The reviewers and I liked your manuscript. Please address the points identified and correct the typos and the paper should be a very nice publication in PLOS computational biology. Therefore, we invite you to submit a revised version of the manuscript that addresses the points raised during the review process.

Please submit your revised manuscript within 30 days Apr 19 2025 11:59PM. If you will need more time than this to complete your revisions, please reply to this message or contact the journal office at ploscompbiol@plos.org. Please include the following items when submitting your revised manuscript:

We look forward to receiving your revised manuscript.

Kind regards,

Rustom Antia

Academic Editor

PLOS Computational Biology

Dominik Wodarz

Section Editor

PLOS Computational Biology

**Additional Editor Comments :**

The reviewers and I liked your manuscript. Please address the points identified and correct the typos and the paper should be a very nice publication in PLOS computational biology.

**Journal Requirements:**

2) We noticed that you used the phrase 'data not shown' in the manuscript. We do not allow these references, as the PLOS data access policy requires that all data be either published with the manuscript or made available in a publicly accessible database. Please amend the supplementary material to include the referenced data or remove the references.

4) We have noticed that you have uploaded Supporting Information files, but you have not included a complete list of legends. Please add a full list of legends for  this Supporting Information file(EEEV_Model.zip) after the references list.

5) We notice that your supplementary Figures are included in the manuscript file. Please remove them and upload them with the file type 'Supporting Information'. Please ensure that each Supporting Information file has a legend listed in the manuscript after the references list.

Potential Copyright Issues:

i) Figure 6C. Please confirm whether you drew the images / clip-art within the figure panels by hand. If you did not draw the images, please provide (a) a link to the source of the images or icons and their license / terms of use; or (b) written permission from the copyright holder to publish the images or icons under our CC BY 4.0 license. Alternatively, you may replace the images with open source alternatives. See these open source resources you may use to replace images / clip-art:

7) Please amend your detailed Financial Disclosure statement. This is published with the article. It must therefore be completed in full sentences and contain the exact wording you wish to be published.

2) If any authors received a salary from any of your funders, please state which authors and which funders.

8) Please ensure that the funders and grant numbers match between the Financial Disclosure field and the Funding Information tab in your submission form. Note that the funders must be provided in the same order in both places as well. Currently, this grant "UC7AI180311" is missing from the Funding Information tab.

9) Please ensure that the Data Availability Statement included in the manuscript is the same as the one provided in the online submission form.

**Reviewers' comments:**

Reviewer's Responses to Questions

Reviewer #1: Overall this manuscript was well-written and easy to understand with clear conclusions from the modeling efforts. I find this work to be potentially impactful and to be nearly ready to publish. I have a number of minor typographical suggestions and just two minor questions/concerns to address to further polish the manuscript. I have no major concerns.

Minor concerns/questions:

1. The use of weights from the linear regression as the sensitivity coefficients is the least clear part of the manuscript. In the methods, a sample equation would aid in clarity to ensure that others could reproduce this technique. Alternatively, the equation could be inserted at line 386. I’ve seen a lot of linear regression and a lot of local and global sensitivity analyses methods, but I am not familiar with this approach of combining the two.

2. In the text, the references included in Table S1 are mentioned. Are these numerous sources cited in the main text as well? If only in the supplemental spreadsheet, then the sources won’t be linked to your paper online and the authors of the work that you used to support aspects of your work won’t get increases to their citations through the online indexing services. My suggestion is to make sure that everything you’ve cited in your supplemental materials is also cited somewhere in the main text.

Suggestions for improving text:

1. In the 2nd sentence of the author summary, “in humans” is used twice.

2. When acronyms are defined or viruses named, only the proper nouns need to be capitalized. This is correct in some places (abstract), but incorrect in several other instances throughout the manuscript for Eastern equine encephalitis, hepatitis C, Venezuelan equine encephalitis, Latin hypercube sampling, Markov chain Monte Carlo, preconditioned Monte Carlo, sequential Monte Carlo with normalizing flows.

3. Sometimes inconsistent use of the Oxford comma. E.g., line 362 after B (orange) is missing a comma and Fig 5 caption after (TC:sg).

4. Lines 113-119 contain one continuous long sentence. While it is grammatically correct, it is recommended to break into two sentences starting from the clause that starts with “which undergoes…” at line 114.

5. Line 222: insert “with” after “consistent”.

6. Line 226: no period at the end of the heading.

7. Line 229: redundant word “that that”.

8. Line 285: insert comma after “polyproteins”.

9. Line 300: insert comma before “respectively”.

10. Line 301: insert comma after “depletion”.

11. Fig 4 caption: Remove (B) before “Parameter scans” and “in panel B” after “red line”.

12. Fig 4C: I understand that the probability density has the same x-axis as the diagram, but it doesn’t share either of the two labeled y-axes. That makes this panel a bit confusing. Perhaps just having the density on a separate panel as you did in e.g., Fig 6 D-F would be clearer.

13. Fig 5 caption: no space in 2,500.

14. Fig 6 is overall very clear. However, panel B is less clear than the rest. Could the cluster labels be added to the groups in panel B?

15. In the Fig 7 caption, parenthesis around the numbered steps would be clearer than the periods that occur midsentences now. E.g., “viral production, 1) 10,000 parameter setps are generated and 2) subsequently simulated.”

16. Line 499: the text that follows the colon for the two-pronged approach includes three items (shutdown, translation, recruitment) and can be read with different interpretations. So it’d be helpful to be explicit to readers about how you want these grouped by including numbered items. I think you meant “two-pronged approach: (1) shut down of host transcription and translation and (2) recruitment …”

17. In the equation/unit conversion that follows line 788, there is an unnecessary ratio of 1 mol / 1 mol.

18. Line 764: 28 parameters are listed here. To aid in clarify, I suggest including as you did in the Fig S5 caption that there are 26 model parameters and 2 scaling factors.

19. Line 789: paragraph should not be indented.

20. Line 790: space between the number of the volume units. Also mu shouldn’t be italicized. (If using latex, this can be accomplished with in the upgreek package.)

21. Line 808: i.e. should be followed by a comma

22. Line 824: no space in the number 10,000.

23. Line 841: cells doesn’t need to be capitalized.

24. Line 842: comma after kidney is not needed before defining the acronym (BHK).

25. Line 879: Is capital T = 0 time? If so, lowercase t is used earlier, so it’d be clearer to keep the notation consistent.

26. Several instances of starting a decimal number with the decimal instead of having a leading 0 before the decimal, throughout the methods. E.g., on line 896, .2 mL instead of the more standard 0.2 mL with the leading 0.

27. The RT-qPCR assay section and following in the methods doesn’t have any indentation, while earlier section indent paragraphs after the first.

28. S1 Fig has 29 panels for 28 distributions. What is in the first panel? I can’t easily find what log(C) is.

29. S2 Fig caption: Insert comma before respectively.

30. S5 Fig caption: Remove space in 2,500.

31. S5 fig and S6 fig are not cross-referenced in the text.

Reviewer #2: This paper presents a computational analysis of Eastern Equine Encephalitis Virus (EEEV), focusing on modeling its intracellular replication. The authors use a rule-based model describing the live cycle of EEEV from attachment to export of virus (and VLP) in mammalian cells. Bayesian inference was used to calibrate the model and reduce uncertainty. The model describes the experimental data well and predicts the important role of ribosomes in viral and production and the EEEV life cycle in general. The study addresses an important topic in virology given EEEVs high mortality rate and the lack of available treatment options. The paper is well written and presents an interesting approach for such a complex model. However, I have a couple minor comments:

- Does the cell line (BHK-21) you use have any immune response? If yes, please explain why the immune response does not affect the viral dynamics? If no, please mention that.

- One model assumption is that ribosomes are within replication spherule? Are there any references?

- In Table S1 (rate rules) maybe add a column with the name of the step/process.

- How many parameters are estimated? 26 or 28 (including two scaling factors). You use both numbers (e.g., in lines 154, 541, 764).

- Can you say anything to the estimated values or are model predictions using Bayesian inference entirely qualitative?

- In the author summary, second sentence: you wrote twice “in humans”

- line 477: double “by setting by setting”

- line 707 and 801: in those sentences may be a word missing

Reviewer #3: The authors present a very comprehensive and well-written manuscript describing a mathematical model of the intracellular replication of eastern equine encephalitis virus, which they have calibrated to time-resolved experimental data of the infection kinetics measured in BHK-21 cells. The authors provide a detailed description of their model and the assumptions they make. The model is a rather large model with 36 equations and a similar number of model parameters, raising some questions about model identifiability. Parameter estimation is done using a Bayesian sampling approach, which provides not point estimates, but posterior distributions over model parameters. The model analyses done include a sensitivity analysis and an identifiability analysis based on points sampled from the posterior distribution; sensitivity analysis results reflect essentially findings in other models developed for related (and simpler) RNA viruses such as HCV or Dengue virus. This paper presents the first model for (the more complex) alpha-viruses.

The work presented is very solid work. I do have only a minor questions concerning the MCMC sampling approach used for parameter estimation. The preconditioned monte carlo approach with normalizing flows used by the authors is not well known in computational biology, and i suggest to add some more details. Of importance, I assume this MCMC-based algorithm requires a burn-in phase to ensure the Markov Chain reaches its stationary distribution, or to ensure proper learnign of the normalizing flows. Can the authors comment? A limited analysis is done with different starting points; I believe this point is critical though to ensure convergence of the sampler to the model posterior distribution. Furthermore, the authors appear to use only default values for the parameters of the preconditioned monte carlo algorithm. Can the authors be sure the 2500 points they sample are sufficient to get a reasonable sample from the posterior?

**Have the authors made all data and (if applicable) computational code underlying the findings in their manuscript fully available?**

Reviewer #1: Yes

Reviewer #2: Yes

Reviewer #3: **No: **Data is available "on request" - should be made fully available as supplementary material.

PLOS authors have the option to publish the peer review history of their article (what does this mean?). If published, this will include your full peer review and any attached files.

Reviewer #1: No

Reviewer #2: No

Reviewer #3: No

**Figure resubmission:**
---

## [Editor Report · Decision Letter 1]

22 Apr 2025

Dear Dr. Faeder,

We are pleased to inform you that your manuscript 'A detailed kinetic model of Eastern equine encephalitis virus replication in a susceptible host cell' has been provisionally accepted for publication in PLOS Computational Biology.

Best regards,

Rustom Antia

Academic Editor

PLOS Computational Biology

Dominik Wodarz

Section Editor

PLOS Computational Biology

---

## [Editor Report · Acceptance letter]

PCOMPBIOL-D-24-02237R1

A detailed kinetic model of Eastern equine encephalitis virus replication in a susceptible host cell

Dear Dr Faeder,

I am pleased to inform you that your manuscript has been formally accepted for publication in PLOS Computational Biology. Your manuscript is now with our production department and you will be notified of the publication date in due course.

With kind regards,

Anita Estes
